# Antibiotic hyper-resistance in a class I aminoacyl-tRNA synthetase with altered active site signature motif

A. Brkic[1], M. Leibundgut[2], J. Jablonska[3], V. Zanki [1], Z. Car [1], V. Petrovic Perokovic[1], A. Marsavelski [1], N. Ban [2] ✉ & I. Gruic-Sovulj [1] ✉

Antibiotics target key biological processes that include protein synthesis. Bacteria respond by developing resistance, which increases rapidly due to antibiotics overuse. Mupirocin, a clinically used natural antibiotic, inhibits isoleucyl-tRNA synthetase (IleRS), an enzyme that links isoleucine to its tRNA$^{Ile}$ for protein synthesis. Two IleRSs, mupirocin-sensitive IleRS1 and resistant IleRS2, coexist in bacteria. The latter may also be found in resistant *Staphylococcus aureus* clinical isolates. Here, we describe the structural basis of mupirocin resistance and unravel a mechanism of hyper-resistance evolved by some IleRS2 proteins. We surprisingly find that an up to $10^3$-fold increase in resistance originates from alteration of the HIGH motif, a signature motif of the class I aminoacyl-tRNA synthetases to which IleRSs belong. The structural analysis demonstrates how an altered HIGH motif could be adopted in IleRS2 but not IleRS1, providing insight into an elegant mechanism for coevolution of the key catalytic motif and associated antibiotic resistance.

Protein synthesis is a central cellular process frequently targeted by natural and man-made antibiotics[1,2] that specifically bind to components of the translational machinery. Bacteria, however, frequently develop resistance to antibiotics through mutations or adaptation, making antibiotic resistance an urgent public health problem[3]. Therefore, a better understanding of the resistance mechanisms is of utmost importance.

Aminoacyl-tRNA synthetases (AARSs) play a key role in the fidelity of translation since they catalyze covalent coupling of amino acids to cognate tRNAs[4,5]. Subsequently, aminoacylated tRNAs (AA-tRNAs) bind to the ribosome in a codon-dependent manner and appropriate amino acids are incorporated into the growing polypeptide chain. AARSs catalyze the formation of AA-tRNA in two steps within the same active site (Fig. 1a). The amino acid is first activated by ATP to form an aminoacyl-adenylate (AA-AMP) intermediate, followed by the transfer of the aminoacyl moiety to the tRNA.

AARSs are divided into two Classes, I and II, each characterized by class-dependent catalytic folds[6] and sequence motifs[7]. Class I AARSs

share the nucleotide-binding fold with Rossmann-like topology that belongs to the larger HUP superfamily[8] and two, so-called, signature motifs, the HIGH and KMSKS motifs. The HIGH motif is located at the tip of helix α1 of the conserved catalytic core comprising segments β1-α1-β2-α2-β3 and α3-β4-α4-β5 that are separated by peptide insertions CP 1 and 2 (Supplementary Fig. 1). The KMSKS sequence is located on the flexible loop that follows the β5 strand. Both motifs, highly conserved in Class I AARSs[9], are an integral part of the active site and are essential for ATP binding and stabilization of the transition state for amino acid activation[10–12].

Isoleucyl-tRNA synthetase (IleRS) is a Class I AARS inhibited by the clinically used antibiotic mupirocin (commercial name Bactroban®) that is naturally produced by *Pseudomonas fluorescences*[13], which competes with isoleucine and ATP for binding at the active site[14,15]. However, two types of IleRSs exist in bacteria that, apart from displaying distinct sequences for the C-terminal tRNA anticodon binding domains, also feature different susceptibility of the active site to mupirocin[16,17]. Specifically, type 1 proteins (IleRS1) are strongly

[1]Department of Chemistry, Faculty of Science, University of Zagreb, Horvatovac 102a, 10000 Zagreb, Croatia. [2]Department of Biology, Institute of Molecular Biology and Biophysics, ETH Zürich, 8093 Zürich, Switzerland. [3]Department of Biomolecular Sciences, Weizmann Institute of Science, 7610001 Rehovot, Israel. ✉e-mail: ban@mol.biol.ethz.ch; gruic@chem.pmf.hr

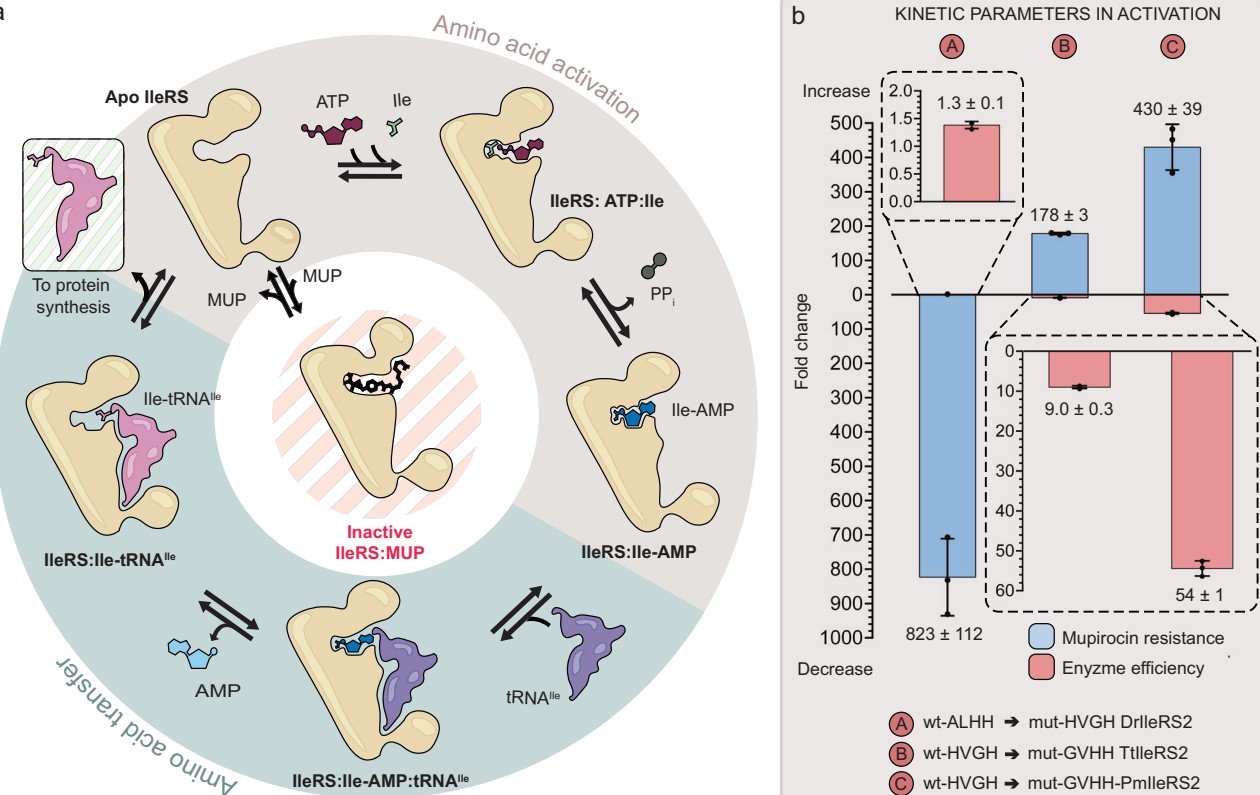

**Fig. 1 | Schematic depiction of the two-step aminoacylation reaction catalyzed by isoleucyl-tRNA synthetase (IleRS). a** The reaction features formation of a reaction intermediate, isoleucyl-adenylate (Ile-AMP), during the amino acid activation step (colored wheat), after which the isoleucyl moiety is transferred to the 2′-hydroxyl group of the tRNA^Ile (colored light green). Bacterial IleRS are susceptible to competitive inhibition by the natural antibiotic mupirocin (MUP)[18]. **b** In IleRS2 the non-canonical version (G/AXHH) of the class I AARS signature motif (HXGH) promotes mupirocin resistance. Exchange of WT non-canonical to canonical (DrIleRS2; A) or WT canonical to non-canonical (TtIleRS2 and PmIleRS2, B and C) signature motifs promotes up to a $10^3$-fold decrease or increase in $K_i$ for mupirocin, respectively, measured at the activation step (blue columns). At the same time, the catalytic efficiency ($k_{cat}/K_M$) of the mutants in the activation step was only slightly compromised (pink columns). "X" in HXGH and G/AXHH indicates a variable position and stands for Ile/Val/Leu/Met/Tyr. Source data are provided as a Source data file. Data are presented as the average value ± SD of three independent experiments.

inhibited by mupirocin ($K_i$ in low nanomolar range[18,19]), while IleRS2, which are homologous to IleRSs from the eukaryote cytosol[16,17], exhibit resistance to mupirocin concentrations that are about three orders of magnitude higher[20]. In some cases, they even reach millimolar levels[17], which we term here hyper-resistance. IleRS1 and IleRS2 generally occur individually in bacteria, but there is a small group of *Bacillaceae* that carry both genes in the genome[21]. IleRS2 plays an important role in providing mupirocin resistance to bacteria in hospital environments, since mupirocin-resistant *Staphylococcus aureus* isolates acquired the *ileS2* gene on a plasmid[22]. Comparison of the crystal structures of *S. aureus* IleRS1 bound to mupirocin and tRNA[14] and *Thermus thermophilus* IleRS2 bound to mupirocin only[15], provides some indications of why in type 2 IleRS the affinity for mupirocin is reduced. However, the comparison is difficult since the structures include different ligands.

Using a combination of phylogenetic, biochemical and X-ray structural analysis we investigated the basis for mupirocin resistance and the origin of hyper-resistance in IleRS2. We found that some IleRS2 harbor an altered Class I HXGH signature motif (with X representing a hydrophobic residue) such that the first and the third amino acids are swapped. This GXHH altered signature motif conveys IleRS2 with hyper-resistance to mupirocin, while catalytic activity is only mildly affected. We determined structures from *Priestia (Bacillus) megaterium*[23] wild-type IleRS1 and IleRS2, both carrying the canonical signature motif, as well as mutants with a swapped GXHH motif,

complexed to an aminoacyl-adenylate analog or mupirocin. These findings revealed why the altered HXGH motif could not be introduced in IleRS1 without abolishing catalysis. Our results provide important insights into the mechanism of antibiotic hyper-resistance and evolution of tRNA synthetases under selective pressure.

## Results

### Mupirocin hyper-resistance is related to the non-canonical class I signature motif

To investigate whether sequence analysis may shed light on the origin of hyper-resistance occurring in some type 2 IleRSs, the sequences of 379 IleRSs were retrieved from representative prokaryotic proteomes[24], aligned, and the phylogenetic tree was inferred. We found that IleRS2 grouped in two distinct clades (Fig. 2a), each, surprisingly, containing a number of IleRS2 lacking the canonical signature HIGH motif (we refer to it as HXGH, as various hydrophobic amino acids are found at the second position, Fig. 2b). Instead, their motifs (dubbed non-canonical) have His at the third position, while the first position is occupied by Gly or rarely Ala. Therefore, this non-canonical motif (G/AXHH), which was not recognized before this study, in essence, has the first and the third position swapped. The exchange is not trivial; the first His stabilizes the transition state of the amino acid activation step, and its mutation in several class I AARSs decreases the corresponding rate by a $10^3$-fold[10,12], while Gly is strongly

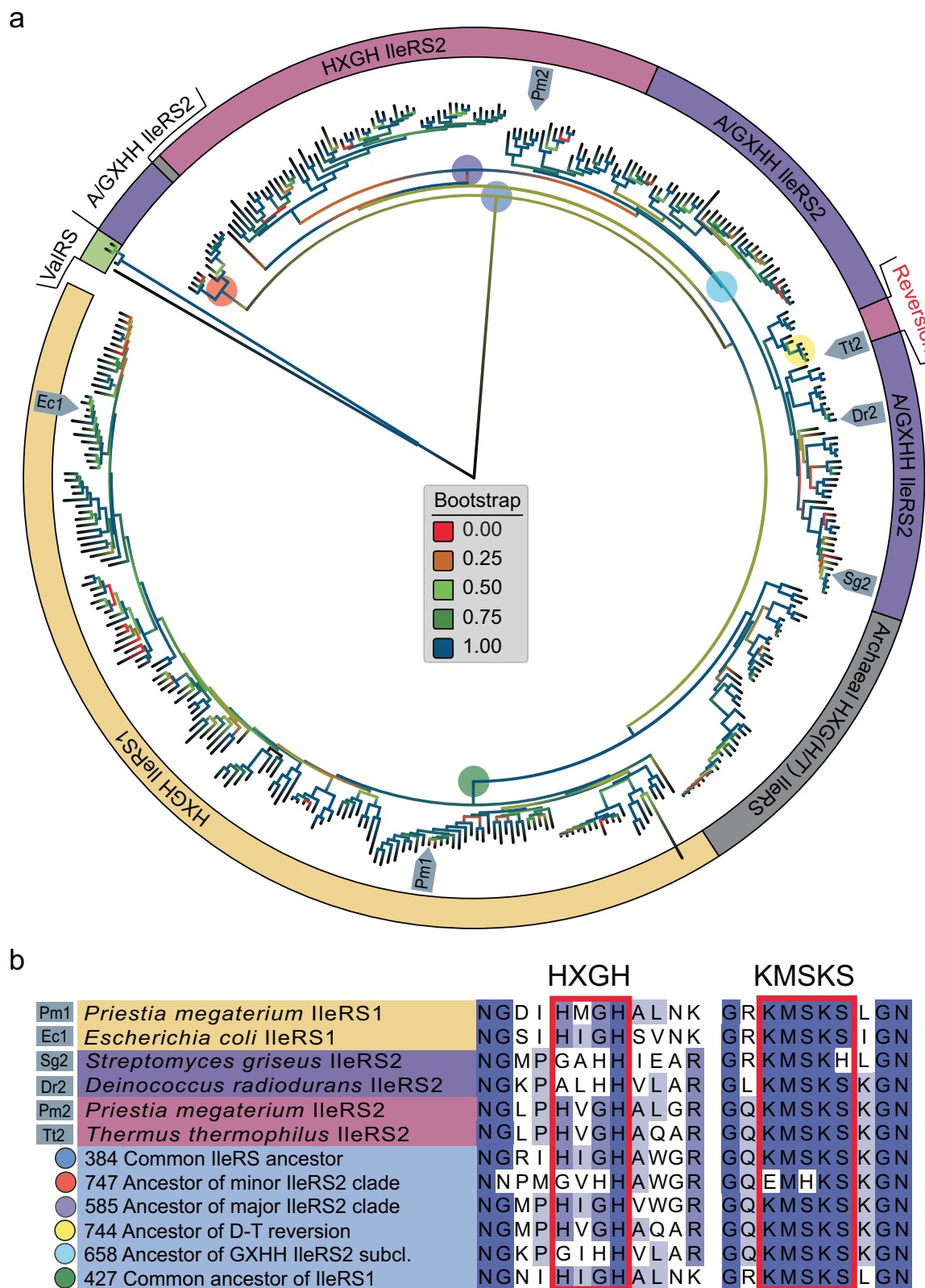

conserved to allow accommodation of the adenine base. We realized that IleRS2 from *Streptomyces griseus* (SgIleRS2), which exhibits a $K_i$ for mupirocin in mM range[17], harbors the same non-canonical motif. Thus, we wondered whether this highly unexpected change in the class I signature motif could be related to mupirocin hyper-resistance.

To explore the mechanism of (hyper-) resistance in greater depth, we tested the mupirocin susceptibility of *Deinococcus radiodurans*

IleRS2 (DrIleRS2), which also carries a non-canonical motif naturally, in this case ALHH. A classical competitive inhibition with respect to both Ile and ATP was observed in the amino acid activation step (Table 1, Supplementary Table 1, and Supplementary Fig. 2). The measured $K_i$ was 6.6 mM, which is more than a $10^3$-fold higher than the $K_i$ of the IleRS2 enzymes carrying the canonical HXGH motif (Table 1) and similar to the SgIleRS2 with the non-canonical motif[17]. Next, we

**Fig. 2 | Phylogenetic analysis of prokaryotic IleRSs. a** The tree was constructed from a multiple sequence alignment of the enzyme's PFAM domains using 379 IleRS sequences (158 HXGH IleRS1, 41 archeal IleRS, 77 HXGH IleRS2, and 86 G/AXHH IleRS2). Source data (the tree and the alignment) are provided as a Source data file. The HXGH motif is present in all IleRS1. IleRS2 can accommodate either HXGH or non-canonical G/AXHH. Naturally occurring reversion of the non-canonical to the canonical motif occurred in the *Deinococcus-Thermus* clade. ValRS sequences were used as an outgroup. A low bootstrap of the IleRS2 early branch is likely a

consequence of a small number of available sequences from *Plantomycetota* and *Chlorflexi* phyla. The IleRS enzymes used in this study are marked on the tree. The key ancestral nodes are depicted by circles in various colors. **b** Sequence alignment of the HXGH and KMSKS signature motifs (within red rectangles) for selected IleRS1 and IleRS2 enzymes and the key ancestral nodes in IleRS evolution. The numbers describe the position of the node in the ancestral tree presented in Supplementary Fig. 4 (source data (the ancestral tree and the alignment of the ancestral nodes) are provided as a Source data file).

**Table 1 | Steady-state amino acid activation parameters and mupirocin inhibition constants[a]**

| Enzyme | Signature motif | $k_{cat}$/s$^{-1}$ | $K_{M,Ile}$/μM | $K_{M,ATP}$/μM | $K_i$ (MUP)$_{Ile}$/μM |
|---|---|---|---|---|---|
| wt-ALHH DrIleRS2 | Natural non-canonical | 45.5 ± 0.6 | 44 ± 2 | 620 ± 20 | 6600 ± 400 |
| mut-HVGH DrIleRS2 | Mutated to canonical | 24.4 ± 0.2 | 17.9 ± 0.5 | 390 ± 10 | 8.1 ± 0.3 |
| wt-HVGH TtIleRS2 | Natural canonical | 27.9 ± 0.4 | 18.7 ± 0.6 | 2200 ± 100 | 0.20 ± 0.01 |
| mut-GVHH TtIleRS2 | Mutated to non-canonical | 7.0 ± 0.1 | 42 ± 1 | 3300 ± 200 | 35 ± 2 |
| wt-HVGH PmIleRS2[b] | Natural canonical | 66 ± 2 | 49 ± 2 | 1600 ± 100 | 1.08 ± 0.06 |
| mut-GVHH PmIleRS2 | Mutated to non-canonical | 6.2 ± 0.1 | 237 ± 6 | 3400 ± 100 | 460 ± 10 |
| wt-HMGH PmIleRS1[b, c] | Natural canonical | 30 ± 2 | 2.1 ± 0.3 | 395 ± 32 | 0.00029 ± 0.00002 |

The values represent the average value ± SD of three independent experiments. The plots are given in Supplementary Fig. 2.
[a]Measured using ATP/PP$_i$ exchange assay.
[b]Values from ref. 21.
[c]Slow tight binding inhibitor.

exchanged the natural non-canonical motif of DrIleRS2 with the canonical (ALHH with HVGH). The mutant, mut-HVGH-DrIleRS2, exhibited an 823-fold drop in $K_i$, corresponding to a drastic loss of resistance, thus linking the non-canonical motif directly with hyper-resistance (Fig. 1b and Table 1). We questioned whether the motif exchange (i.e., the swap of the first and the third motif position) will increase the resistance of the canonical IleRS2. To address this, we chose two IleRSs from species where the canonical HXGH is present, *T. thermophilus* IleRS2 (TtIleRS2) and *P. megaterium* IleRS2 (PmIleRS2), and exchanged their canonical HVGH motif with GVHH. Both mutants (mut-GVHH-TtIleRS2 and mut-GVHH-PmIleRS2) indeed experienced around 200-fold increase in $K_i$ (Fig. 1b and Table 1). Altogether, our data provide compelling evidence that the non-canonical form (G/AXHH) of the Class I AARS signature motif is responsible for up to a 10$^3$-fold increase in mupirocin resistance in IleRS2.

## The non-canonical motif cannot be functionally accommodated in IleRS1

Alteration of the HXGH motif in some IleRS2 was highly surprising considering the key catalytic role of this motif in the amino acid activation. Hence, we explored how this is accomplished, and whether it comes at a trade-off with IleRS activity, using kinetics and structural approaches.

As shown in Table 1, DrIleRS2, which naturally carries the non-canonical motif, shares catalytic efficiency in the activation step ($k_{cat}/K_M$) with the canonical PmIleRS2 and TtIleRS2, supporting a lack of catalytic trade-offs related to hyper-resistance. Introducing the non-canonical GVHH motif into PmIleRS2 and TtIleRS2 also did not strongly affect the enzymes, yet hyper-resistance, in this case, came at the expense of increased $K_M$ and decreased $k_{cat}$ values of up to 10-fold (Table 1 and Fig. 1b). Finding that the non-canonical motif is well tolerated in IleRS2 is consistent with its broad distribution among IleRS2 (Fig. 2).

In sharp contrast, the phylogenetic analysis did not identify a single IleRS1 with the non-canonical motif (Fig. 2), strongly suggesting that HXGH motif variations are not tolerated among IleRS1. To test this, we exchanged the natural HXGH motif in IleRS1 from *P. megaterium* and *Escherichia coli* with GXHH. GXHH-mutants showed a lack of product formation during prolonged reaction times even at 15 μM

enzyme (Supplementary Fig. 3), confirming that the active site of IleRS1 cannot productively accommodate the non-canonical motif.

To reconstruct the evolutionary origin of the non-canonical motif, we inferred the IleRS ancestral states from our phylogenetic analysis (Fig. 2b and Supplementary Fig. 4). The HXGH motif is found in most of the earliest ancestors, i.e., the common ancestor of all IleRSs (node 384), the IleRS1 ancestor (node 427) and the ancestor of the major IleRS2 clade (node 585). The exception is the ancestor of the minor IleRS2 clade (node 747), which harbors the non-canonical GXHH motif. Among the inner nodes, GXHH can be found at node 658, which represents the GXHH subclade in the major IleRS2 clade. The presence of GXHH in two distant IleRS2 nodes indicates that the non-canonical motif was acquired at least twice during evolution of IleRS2. That the motif exchange is not detrimental to IleRS2 is also supported by a natural reversion of the non-canonical to the canonical motif in the *Deinococcus-Thermus* clade (node 744, ancestor has the HXGH motif). That said, laboratory exchange of HVGH back to GVHH in TtIleRS came at a minor expense of its catalytic efficiency (Table 1). The observation that all early ancestors (except node 747) carry the class I AARS-dominating HXGH motif may indicate that both IleRS2 and IleRS1 emerged with the HXGH motif. Mutation of the IleRS2 signature motif appears later in the evolution, presumably under selective pressure to withstand higher antibiotic concentrations in the environment.

## Structural basis of the non-canonical motif accommodation in IleRS2

To understand how IleRS2, but not IleRS1, accommodates the non-canonical motif, we used X-ray crystallography to determine the structures of a pair of IleRS enzymes from the same organism. Apart from wild-type PmIleRS1 and PmIleRS2, both of which harbor the canonical HXGH motif, we also structurally characterized their corresponding GXHH mutants, all in complex with a non-hydrolyzable analog of the isoleucyl-adenylate reaction intermediate, Ile-AMS (Supplementary Table 2 and Supplementary Fig. 5).

The overall fold of the full-length wild-type PmIleRS1 and PmIleRS2 (Fig. 3a, b) and the topology and architecture of their active sites (Supplementary Fig. 1) correspond well to the previously determined structures of IleRS1 from *S. aureus*[14] and IleRS2 from *T. thermophilus*, *Candida albicans*, and *Saccharomyces cerevisiae*[15,25,26]. Additionally, the

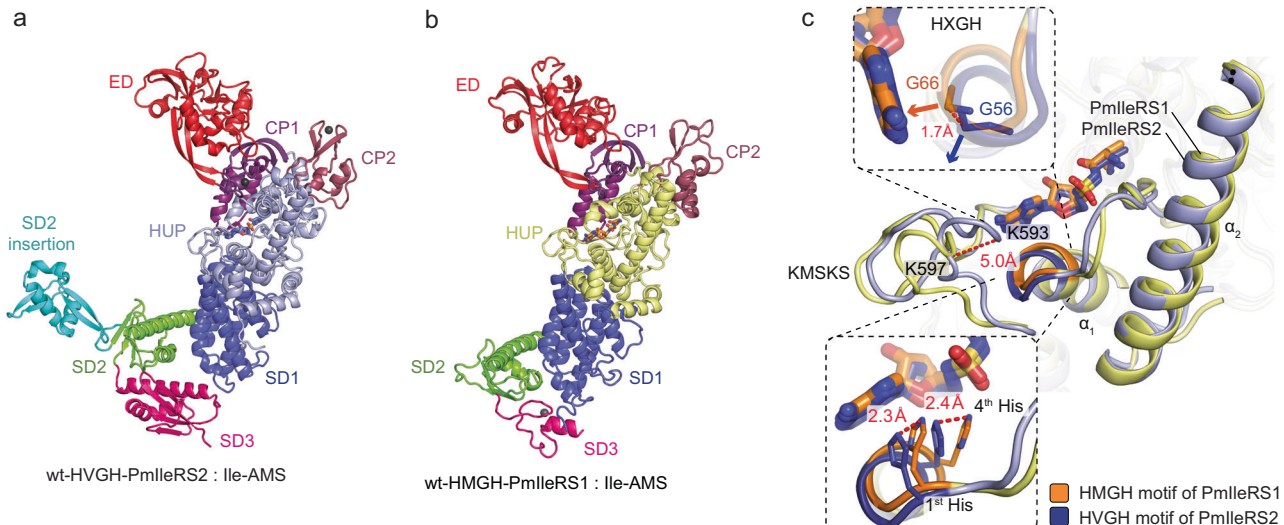

**Fig. 3 | Comparison of crystal structures of PmIleRS2 and PmIleRS1 in complex with the non-hydrolysable analog of the reaction intermediate Ile-AMS.** **a, b** The canonical structures with the HUP catalytic domain (colored gray in IleRS2 and yellow in IleRS1), the CP1 and CP2 domains (CP refers to connective peptide, colored purple and raspberry, respectively) and the editing domains (ED, colored red) inserted into CP1 are visualized. The full-length C-terminal domain of type 2 IleRS is resolved, revealing three subdomains (SD1-SD3, colored blue, green and pink, respectively), among which SD3 differs in size, fold and lack of the zinc-binding motif relative to SD3 in IleRS1. An insertion (colored cyan) into SD2 is observed in IleRS2. For details see Supplementary Fig. 6. **c** Structural overlay of the IleRS1 and IleRS2 HUP catalytic cores (residues 50–174 and 523–635 in IleRS1 and 40–168 and 513–632 in IleRS2) bound to Ile-AMS revealed overlapping positions of Ile-AMS and a conformational rearrangement of the active site in IleRS2 relative to IleRS1. The tip of helix α1 comprising the HXGH motif (blue in IleRS2 and orange in IleRS1) and the KMSKS loop move towards each other in IleRS2. This repositions both the first and fourth histidine residues as well as the glycine α-carbon (insets).

structure of the C-terminal tRNA anticodon binding domain, which differs among the two IleRS types[16], is resolved for a type 2 protein (Fig. 3a and Supplementary Fig. 6). The structures reveal that the reaction intermediate analog Ile-AMS in both wild-type enzymes binds to the active site in a canonical manner[15] (Supplementary Fig. 7). However, superimposing the HUP catalytic cores of IleRS1 (residues 50–174 and 523–635) and IleRS2 (residues 40–168 and 513–632) unraveled conformational rearrangements within the signature motifs in the active sites (Fig. 3c). Relative to IleRS1, the KMSKS loop assumed a more closed conformation in IleRS2, as indicated by a 5.0 Å displacement of the first Lys backbone towards helix α1 (Fig. 3c). While the KMSKS loop is known to be flexible and to change its conformation during the reaction[27], we also found that the HXGH motif in IleRS2 is displaced towards the KMSKS loop (Fig. 3c). The latter affects repositioning of the first and the fourth histidine of the HXGH motif by 2.3 and 2.4 Å, respectively, and the Gly α-carbon by 1.7 Å relative to the analogous positions in IleRS1 (Fig. 3c, insets). The displacement (shift) of the HIGH motif in IleRS2 is of utmost importance for the accommodation of the non-canonical A/GXHH motif and emergence of hyper-resistance, as discussed below. Hence, to explore it further, we extensively analyzed the residues in the immediate vicinity of the HXGH motif using mutagenesis combined with biochemical and structural characterization (exemplified for the W130Q-PmIleRS2 mutant in Supplementary Fig. 8 and Supplementary Table 2). However, no simple mechanism responsible for the observed conformational change that remodels the IleRS2 active site emerged, suggesting that the shift of the HXGH motif is more complex and likely deeply rooted in the architecture of the IleRS2 active site.

How does alteration of the canonical motif (i.e., exchange of the first and the third position in HXGH) influence the PmIleRS structures and, hence, their function? At the level of the overall fold, the WT enzymes and corresponding GXHH mutants are highly superimposable (Supplementary Fig. 9a, d). However, while the binding of Ile-AMS to the active sites of both GXHH mutants parallels its binding to the corresponding wild-type enzymes (Fig. 4, Supplementary Fig. 7),

some distinctions that are mainly related to the precise accommodation of the adenine base emerged. In PmIleRS1, introducing the GMHH mutation leads to a 1.5 Å shift of the adenine base and a 2.1 Å move of Phe586, which loses its stacking with the adenine moiety (Fig. 4a). Such a distorted geometry of the active site likely results in a non-productive binding of the ATP substrate and concomitant loss of mut-GMHH-IleRS1 activity (Supplementary Fig. 3). Such a scenario does not occur in PmIleRS2, where the tip of the α1 helix in both the mut-GVHH-PmIleRS2 and corresponding wild-type enzyme are similarly rearranged (shifted) compared to IleRS1, resulting in a 1.7 Å displacement of the backbone in the third position of the motif (Fig. 3c, inset). This allows accommodation of the third His in mut-GVHH-PmIleRS2 without displacement of the adenine moiety. This way, the third His can take over the role of the first His in the canonical reaction mechanism (Fig. 4b).

**Structural basis of mupirocin resistance**

The non-canonical motif endows IleRS2 with hyper-resistance, the structural basis of which cannot be assessed by crystallography due to a $K_i$ in the mM range. Therefore, to deepen our understanding of mupirocin resistance in general and to infer the mechanism of hyper-resistance, we determined the crystal structures of wild-type PmIleRS1 (sensitive) and PmIleRS2 (resistant), both of which carry the canonical HXGH motif and are amenable in mupirocin-bound form when the inhibitor is supplied at sufficiently high concentrations (Fig. 5, Supplementary Table 2, and Supplementary Fig. 10). The overall structures of mupirocin-bound PmIleRS1 and PmIleRS2 overlap well with those determined in complex with Ile-AMS (Supplementary Fig. 9c, e). In both enzymes, mupirocin binds to the active site mimicking the interaction intermediate Ile-AMP (Supplementary Fig. 9c, e), as expected based on the previous structural data[14,15] and the competitive mode of inhibition (Table 1). The binding of mupirocin to both (sensitive and resistant) enzymes (Supplementary Fig. 11) is similar as described[15]; however, we are able to reveal distinct interactions. We observe the backbone of Pro56 and the side chain of Asn70 forming interactions with O13 and O7 atoms of mupirocin, respectively, in

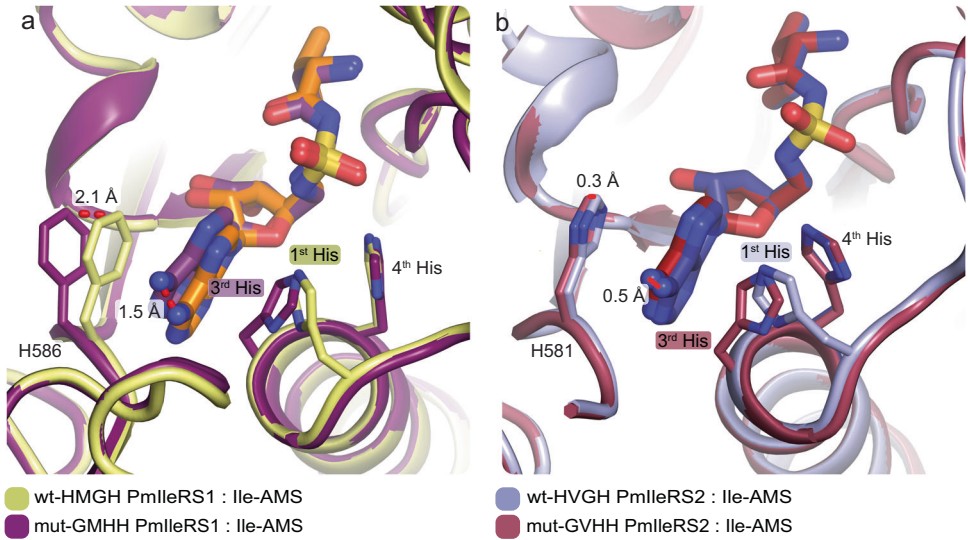

wt-HMGH PmIleRS1 : Ile-AMS
mut-GMHH PmIleRS1 : Ile-AMS

wt-HVGH PmIleRS2 : Ile-AMS
mut-GVHH PmIleRS2 : Ile-AMS

**Fig. 4 | Structural overlay of the active sites of WT PmIleRS1 and PmIleRS2 with mutants where histidines in the signature motif are exchanged. a** In IleRS1, the third His from GV**H**H promotes mispositioning of the adenine base that contributes to the abolished catalysis. **b** In IleRS2, the general shift of the GMHH motif relative to IleRS1 allows accommodation of a His in the third position without mispositioning of the adenine moiety. In both the mutated (mut-) and wt-structures, the third His HNε2 from GXHH adopts an equivalent position as the catalytically relevant HNε2 of the first His from HXGH.

PmIleRS1, which are absent in PmIleRS2 (Fig. 5b, d). Furthermore, stacking interactions of Phe586 and the unsaturated C2-C3 bond of the monic acid A part (Fig. 5a) contributes to the affinity in IleRS1 (Fig. 5b), while the analogous His582 is reoriented in PmIleRS2 (Fig. 5d). Finally, we found that the carboxylate oxygen of the nonanoic acid part of mupirocin (Fig. 5a) establishes an H-bond with the backbone NH of K597 from the KMS**K**S loop only in PmIleRS1 (Fig. 5b, d). It was demonstrated[28,29] that H-bonding to a charged group may contribute 3-4 kcal/mol to binding, which translates into up to a $10^3$ contribution to a binding constant. Indeed, H-bonding of the tyrosyl-tRNA synthetase active site aspartate and the hydroxyl group of the tyrosine substrate is estimated to contribute a $10^3$-fold to the selectivity of the enzyme against phenylalanine[28]. In PmIleRS2, the closed conformation of the KMSKS loop (Fig. 5d) forces binding of the nonanoic carboxylate such that it is positioned at 4.1 Å distance to the NH of Lys593 (KMS**K**S). It is this lack of carboxylate stabilization by H-bonding that may strongly affect the IleRS2 affinity towards mupirocin and provide resistance.

Interestingly, we observed a kink of helix α2 (from residues 123–126) solely in the crystal structure of mupirocin-bound PmIleRS2 (Supplementary Fig. 12a). To explore whether these structural changes in PmIleRS2 are due to different conformational readjustments of the active site upon ligand binding (i.e., induced fit)[30] or sampling of different enzyme's conformations from solution (i.e., conformational selection)[31], we aimed to crystallize IleRS2 in the apo form. Because this approach remained unsuccessful, we used molecular dynamics (MD) to address the conformation of the helix α2 in the absence of ligands. When we removed the ligands from the PmIleRS2 structures, we found that independent of whether we started from the apo structure with the regular helix α2 (obtained by removing Ile-AMS) or the distorted one (obtained by removing mupirocin), helix α2 predominantly (>99%) went into the distorted conformation during 360 ns simulations (Supplementary Fig. 12b). Noteworthy, an irregular helix α2 structure was also observed in MD simulations of IleRS2 bound to Ile-AMS, yet here distortion was observed after around 180 ns (Supplementary Fig. 13). Thus, the MD data suggest that the kink in helix α2 is part of the conformational flexibility of IleRS2 and is neither induced by mupirocin binding nor by different packing environments in the different space groups that PmIleRS2:mupirocin and PmIleRS2:Ile-AMS crystallized in (Supplementary Table 2). Instead, it appears that Ile-AMS and

mupirocin sample different IleRS2 conformations that exist in solutions.

### Modeling of hyper-resistance in PmIleRS2

Structural data unraveled that in the PmIleRS2:mupirocin complex, the tip of the helix α1 carrying the HXGH motif is shifted back towards helix α2, resulting in 2.9 or 3.3 Å displacements of the first or fourth His, respectively, relative to PmIleRS2:Ile-AMS (Fig. 5e). No such ligand-dependent rearrangements in the HXGH motif are observed in PmIleRS1 (Fig. 5c). Thus, in the PmIleRS2:mupirocin complex the HXGH motif adopts the same position as in the IleRS1 structures (Supplementary Fig. 14a). The back-shift of the HIGH motif is likely promoted by binding of the nonanoic part of mupirocin to the channel between the HIGH motif and the KMSKS loop forced by the closed conformation of the KMSKS loop in PmIleRS2 (Supplementary Fig. 14b).

We hypothesize that this back-shift, which places the third amino acid of the HXGH motif into close proximity to the adenine base, provides the basis of A/GXHH motif-promoted hyper-resistance. To address this hypothesis, we in silico exchanged the canonical motif in the structure of wt-HVGH-PmIleRS2 bound to mupirocin with the GVHH sequence. The coordinates for the latter were taken from the mut-GVHH-PmIleRS2:Ile-AMS structure. The resulting clash due to a His in the third position with both, the nonanoic part and pyrrole ring of mupirocin (Fig. 6), likely explains how the presence of the non-canonical motif prevents mupirocin from binding and leads to the further substantial increase in $K_i$.

## Discussion

Translation is a central cellular process and a frequent target for antibiotic action. Most of the known translation-related antibiotics act on ribosomes[32]. A smaller number target AARSs, the essential enzymes that secure correctly aminoacylated tRNAs[2]. Among them, the best-known is mupirocin, which blocks IleRS by binding to both the isoleucine and ATP binding pockets[15] and is primarily active against gram-positive pathogens, including methicillin-resistant *S. aureus* (MRSA)[33,34].

In bacteria, two IleRS types, IleRS1 and IleRS2, may perform the housekeeping function. Mupirocin strongly inhibits IleRS1 and modestly IleRS2, with the difference in $K_i$ ranging from $10^3$–$10^5$-fold. Accordingly, the presence of IleRS2 as a sole[17] or a second[21] gene in the

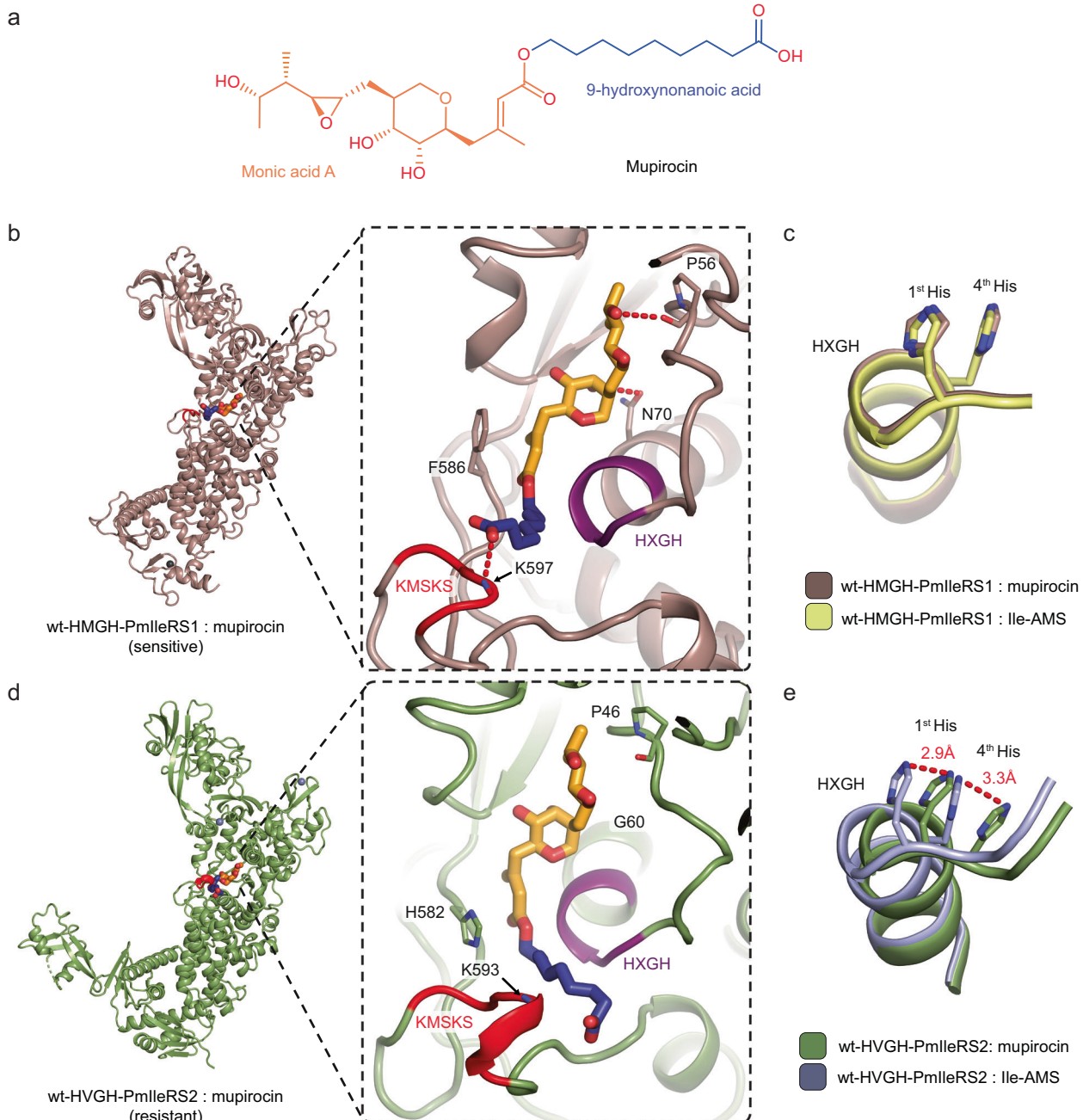

**Fig. 5 | Structures of PmIleRS1 and PmIleRS2 bound to mupirocin. a** Mupirocin is an ester of monic acid A and 9-hydroxynonanoic acid. Within the active sites of IleRS1 and IleRS2, the monic acid part mimics the Ile-AMP interactions, whilst the nonanoic acid part is oriented towards the KMSKS loop. **b**, **c** In IleRS1, unique interactions with mupirocin include the carboxyl group of the nonanoic acid moiety, which establishes an H-bond with the backbone NH of K597 in the KMSKS loop, the hydrogen bonds between the monic acid part and a proline and an asparagine and stacking interactions of phenylalanine. All interactions are depicted

in Supplementary Fig. 11. The position of the HXGH motif remains unaltered upon mupirocin binding. **d**, **e** In the IleRS2 active site, mupirocin binds via a lower number of interactions, providing the basis for resistance. Stabilization of the nonanoic carboxylate by H-bonding is precluded by a closed conformation of the KMSKS loop, thereby forcing the nonanoic acid moiety into the cleft between the HXGH motif and the KMSKS loop. As a result, the HXGH motif is pushed away from the KMSKS loop.

genome or on a plasmid[22] confers the resistance. Antibiotic resistance is a major public health problem worldwide[3] and resolving the underlying mechanisms by which it evolves and operates broadens our capacity to tackle it.

The mechanism of mupirocin resistance has not yet been fully understood. The structural basis was indecisive because a comparison of the mupirocin-bound structures of *T. thermophilus* IleRS2[15] and *S. aureus* IleRS1[14] is not straightforward considering that the IleRS1 structure besides mupirocin also contains the tRNA, which may

promote conformational changes of the active site loops. To alleviate this problem, we solved the structures of IleRS1 (sensitive) and IleRS2 (resistant) from *P. megaterium* bound to mupirocin only (Fig. 5). The structures provide an important progress in understanding the resistance mechanism by identifying that the closed conformation of the active site KMSKS loop in IleRS2 precludes H-bonding stabilization of the mupirocin's carboxylate (Fig. 5d and Supplementary Fig. 14b). Because H-bonding to a charged group has been shown as a powerful mechanism to ensure a high level of specificity in molecular

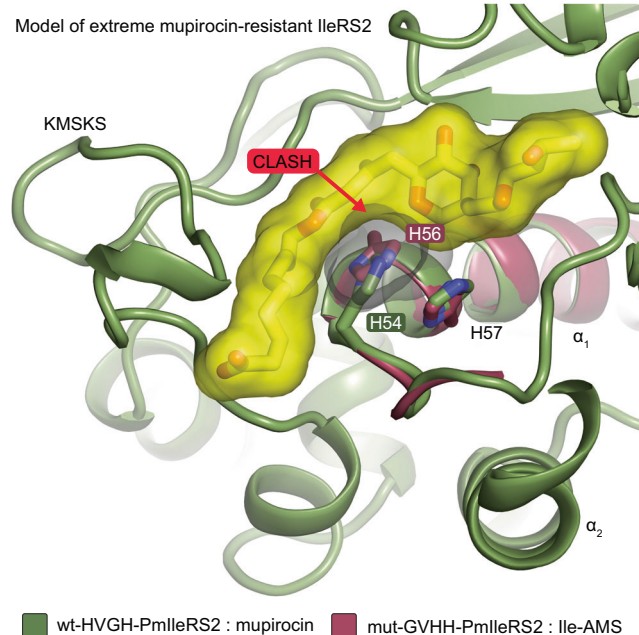

Model of extreme mupirocin-resistant IleRS2

KMSKS

CLASH

H56

H54  H57  α₁

α₂

■ wt-HVGH-PmIleRS2 : mupirocin  ■ mut-GVHH-PmIleRS2 : Ile-AMS

**Fig. 6 | Mupirocin modeled into the active site of mut-GVHH-IleRS2 displaying a hyper-resistance phenotype.** The coordinates of the GVHH motif were taken from the mut-GVHH-PmIleRS2:Ile-AMS structure (colored deep red) and were superimposed onto wt-HVGH-PmIleRS2:mupirocin (colored green). A G56H exchange results in clashes with distances below 2.5 Å excluding hydrogens, consistent with the hyper-resistance observed in IleRS2 variants harboring the non-canonical motif (Table 1).

recognition[29] and may strongly contribute to binding of mupirocin to IleRS1, loss of this interaction in IleRS2 may have been adopted as an efficient resistance mechanism during evolution. In accordance, no H-bonding to the mupirocin's carboxylate was observed in the TtIleRS2:mupirocin structure[15] or a model of mupirocin bound to eukaryotic IleRS2[25]. The structural data are also consistent with the finding that monic acid A alone (Fig. 5a) cannot act as an inhibitor[35].

A possible role of tRNA in modulating the binding of mupirocin seems unlikely, as kinetic data showed that the $K_i$ for mupirocin in the amino acid activation and the two-step tRNA aminoacylation reactions are similar[17,18,36]. Consistent with these findings, no direct interaction between mupirocin and the tRNA was observed in the structure of SaIleRS1 complexed to tRNA and mupirocin (1QU2)[14]. Further, the active site residues contacting mupirocin in our PmIleRS1:mupirocin structure (Fig. 5b) and the SaIleRS1:tRNA:mupirocin complex, including the KMSKS loop and the tip of the α1-helix harboring the HIGH motif, adopt a highly similar arrangement, indicating that their conformation is independent of the presence of tRNA. Nevertheless, based on our structural findings we cannot exclude that upon the binding of the 3′-end of the tRNA to the catalytic site, the KMSKS loop, which displays increased local temperature factors, may become directly or indirectly stabilized or change its conformation[37].

A major surprise, however, came from the finding that hyper-resistance ($K_i$ in mM range) found in some IleRS2 (Table 1 and ref. 17) originate from their naturally altered key catalytic motif, the HXGH motif. We mimic the motif alteration by the exchange of the first and third motif positions in PmIleRS2 that naturally carries the canonical HVGH motif. This GVHH-PmIleRS2 mutant with a bulky histidine, instead of glycine, at the third position displayed around 430-fold increase in $K_i$ for mupirocin (Table 1) due to a severe steric clash of the third histidine with mupirocin (Fig. 6). Motif exchanges in both directions (canonical to non-canonical and vice versa) that we carried

out in several IleRS2 enzymes unambiguously link the non-canonical motif with hyper-resistance.

Mutational modification of the target that diminishes its interaction with an antibiotic is one of the classical ways how resistance develops[3,38]. Yet, we found as remarkable that the modification of IleRS2 includes exchange of the most conserved first and third positions[39] of the class I AARS signature motif without compromising the variant's housekeeping role and its broad distribution in bacteria (Fig. 2). In contrast, no IleRS1 with non-canonical signature motif has been found in nature so far (Fig. 2). Even more, introducing the non-canonical motif in laboratory-produced IleRS1 diminishes its activity (Supplementary Fig. 3). So, the signature motif exchange appears as an extraordinary resistance mechanism adopted exclusively in IleRS2. The insights into why IleRS2, but not IleRS1, can tolerate the non-canonical signature motif came from the crystal structures of the pair of PmIleRS1 and PmIleRS2, both in complex with Ile-AMS. Specifically, in PmIleRS2, both wild-type and the GXHH-mutant, the tip of the helix α1 carrying the HXGH or GXHH motif is rearranged (shifted) and the α-carbon of the third motif residue is displaced (away from the adenosine) by 1.7 Å relative to the analogous position in PmIleRS1 (Fig. 3c). This motif shift in IleRS2 solves the key problem associated with the change of a sequence, i.e., it enables accommodation of a larger amino acid, namely a His instead of the highly conserved Gly[39], in the third position without a clash with the adenine base (Fig. 4b). In PmIleRS1, where no rearrangement of either HXGH or GXHH motif has been observed (Fig. 3c), the histidine at the third position promotes mispositioning of the adenine moiety (Fig. 4a), resulting in the abolished activity of mut-GVHH-PmIleRS1 (Supplementary Fig. 3). That the steric constraints introduced by the third His promote a non-productive binding of ATP, which in turn diminishes IleRS1 activity, is further supported by mutational of analysis of the HLGH motif in methionyl-tRNA synthetase wherein Gly to Pro substitution induces severe $k_{cat}$, but not $K_M$ (ATP), effect in the activation step and a loss of coupling binding energy between the amino acid substrate and ATP[40].

Both *ileS1* and *ileS2* genes are proposed to be of ancient bacterial origin[17] and have presumably been under selective pressure to develop resistance against naturally occurring mupirocin produced by *Pseudomonas fluorescens*[13]. However, did the same selective forces hold for both IleRS types? IleRS1 is found preferentially in faster- while IleRS2 in slower-growing bacteria[21], raising the question of whether a negative selection against catalytic trade-offs more prominently shaped IleRS1. That said, clinical isolates of *S. aureus*, including MRSA strains[34], could evolve only a low resistance by IleRS1 mutations, while bacterial isolates with a higher IleRS1 resistance can be obtained exclusively in the laboratory because of their compromised fitness. Furthermore, some of the laboratory-selected *S. aureus* strains carry, among other mutations, also a His to Gln mutation at the fourth position of the SaIleRS1 HIGH motif. Likely due to reduced catalytic properties, such mutations have not been observed in clinical isolates, explaining in part why, in spite of our considerable effort to engineer functional IleRS1 with a non-canonical motif, no active enzyme was obtained. Taken together, our results support a view that the differences between IleRS1 and IleRS2 are rooted deeply in the overall architecture of the catalytic domain, which is indicative of separate evolution trajectories of the IleRS1 and IleRS2 catalytic folds, likely because of distinct selection forces.

## Methods
### Cloning and mutagenesis
The expression vectors encoding wild-type HIGH-EcIleRS1, HMGH-PmIleRS1 and HVGH-TtIleRS2 are described[21,41]. Expression vectors for wild-type ALHH-DrIleRS2 and HVGH-TtIleRS2 were prepared by inserting the coding sequences into pET28b(+). For ALHH-DrIleRS2, the coding sequence was PCR-amplified from genomic DNA of *D. radiodurans* R1 ATCC 13939 and for wt-HVGH-TtIleRS2 from the

plasmid TEx18A07 from the Riken BRC DNA Bank made available through the National BioResource Project of the MEXT/AMED, Japan. The final constructs, which encoded non-codon optimized full-length enzyme coding sequences (CDS) fused to an N-terminal hexa-histidine tag, were verified by sequencing (Macrogen Inc., commercial service). Single point mutations were introduced by quick-change site directed mutagenesis (Q5-SDM kit (NEB, cat: E0554)) using custom-designed primers (Supplementary Table 3). The custom design features a pair of the short outward-facing (from the mutation target) primers with the mutation introduced at the 5′-ends. When suboptimal codon usage prohibited the mutagenic primer design (e.g. ALHH-DrIleRS2), cassette mutagenesis[42] was used, where the parts of the original CDS were exchanged with synthetic cassettes carrying desired mutations (Twist Bioscience). The mutations were verified by sequencing (Macrogen Inc., commercial service).

## Protein expression and purification
Expression vectors carrying wild-type and mutant IleRSs were either transformed into chemically competent BL21(DE3) *E. coli* cells (Novagen, cat: 69450) or, in the case of TtIleRS2-carrying plasmids, into Rosetta2 (DE3) (Novagen, cat: 71397). For large-scale protein expression, cells were grown to early-log phase (0.4 < OD < 0.6) in LB broth supplemented with 1 mM MgCl₂, which greatly improved the overall protein yield. Protein expression was induced by 0.25 mM IPTG (EcIleRS1, PmIleRS1, PmIleRS2, DrIleRS2 and their mutants) or 1 mM IPTG (Sigma-Aldrich, cat: I6758) (wt- and mut-TtIleRS2) for 3 h at 37 °C (wt and mut-EcIleRS), 5 h at 30 °C (wt and mut-PmIleRS1), or 16 h at 15 °C (DrIleRS2, TtIleRS2, PmIleRS2 and their mutants), followed by harvesting the cells using centrifugation and freezing the pellet at −80 °C. The latter significantly facilitated cell lysis efficiency and protein recovery.

The cell pellet from 0.5 L of culture was thawed at RT and resuspended in 10 ml IMAC A buffer (25 mM Hepes-KOH pH = 7.5 at 20 °C, 500 mM NaCl, 10 mM imidazole, 10 mM 2-mercaptoethanol) followed by addition of 10 µg/ml DNase I (Merck-Millipore, cat: 260913), 10 µg/ml RNase A (Merck-Millipore, cat: 55674), 25 µg/ml lysozyme (Roche, cat: 10837059001) and 0.1 mM PMSF (Sigma-Aldrich, cat: P7626) (final concentrations). The suspension was lysed by sonication (10 times for 45 s on 50 % sonication power with 1-min intervals between the pulses). The lysate was cleared by centrifugation (1 h at 25,000 × *g* at 4 °C), and the supernatant containing soluble proteins was subjected to IMAC purification on a HisTrapHP 5 mL column (Cytiva, cat: 17524802) connected to an AktaPure 25 system (Cytiva). The column was washed with 20 CV of IMAC A and 10 CV IMAC B (25 mM Hepes-KOH pH = 7.5 at 20 °C, 50 mM NaCl, 10 mM imidazole, 10 mM 2-mercaptoethanol) buffers, and the protein of interest was eluted by a linear gradient of imidazole using 20 CV of IMAC C buffer (25 mM Hepes-KOH pH = 7.5 at 20 °C, 50 mM NaCl, 250 mM imidazole, 10 mM 2-mercaptoethanol). The fractions were pooled and the proteins further purified by IEX on a MonoQ 16/10 HR column (Amersham Bioscience, discontinued) equilibrated in IEX buffer A (25 mM Hepes-KOH pH = 7.5 at 20 °C, 50 mM NaCl, 10 mM 2-mercaptoethanol). The unbound proteins were washed away with 20 CV IEX A, and elution of the protein of interest was achieved by a linear gradient of ionic strength using 20 CV IEX buffer B (25 mM Hepes-KOH pH = 7.5 at 20 °C, 1000 mM NaCl, 10 mM 2-mercaptoethanol). The fractions containing pure protein were pooled, and the buffer was exchanged to storage buffer (25 mM Hepes-KOH pH = 7.5 at 20 °C, 50 mM NaCl, 10 mM 2-mercaptoethanol). The proteins were concentrated to 16.5 mg/ml (HVGH-PmIleRS2 and mutants thereof), 20 mg/ml (HMGH-PmIleRS1, HIGH-EcIleRS1 and mutants thereof) or 10 mg/ml (HVGH-TtIleRS2 and mutants thereof), divided into 50 µl aliquots and flash frozen by plunging into liquid nitrogen. The purity of the final preparations is >95% as estimated by SDS-PAGE analysis.

## Determination of kinetic parameters in the activation step
Inhibition constants ($K_i$) for mupirocin towards L-Ile and ATP in the activation step of the aminoacylation reaction catalyzed by wild-type and mutant IleRSs were determined by the ATP-PP$_i$ exchange assay[21,43]. The reaction started by addition of the enzyme and was conducted at 30 °C in 20 µl reaction mixtures containing 55 mM Hepes-NaOH pH = 7.5 (255 mM in case of ALHH-DrIleRS2), 30 mM MgCl₂, 1 mM NaPP$_i$, 5 mM DTT, 0.1 mg/ml BSA, $^{32}$P-PP$_i$ (0.2−0.4 µCi/µmol) (Perkin Elmer, cat: NEX019010MC) and varying amounts of the enzyme, L-Ile, ATP and mupirocin. L-Ile and ATP concentrations ranged from 1/10 $K_M$ to 10 $K_M$ values. When measuring the $K_i$ towards one of the substrates, the other was held in saturation (concentrations at least 10 $K_M$). The aliquots of the reaction mixture (1–2.5 µl) were quenched in 2 volumes of 500 mM NaOAc pH = 4.5, 0.1% SDS, and formed $^{32}$P-ATP was separated from the remaining $^{32}$P-PP$_i$ by thin-layer chromatography on Polygram CEL 300 PEI UV254 TLC plates (Macherey-Nagel, cat: 801063) using 750 mM KH₂PO₄ pH=3.5, 4 M urea buffer. Visualization and quantification of the signals was performed using a Typhoon Phosphoimager 5 (General Electric) and accompanying ImageQuant™ TL software (SV: 10.2). The kinetic constants were determined using a global fit option of the competitive inhibition model of GraphPad Prism 6 software (SV: 6.01). All experiments were performed in triplicates. The $k_{cat}$ was determined by dividing the maximal velocity, $V_M$, with the used enzyme concentration.

## Protein crystallization and crystal stabilization
Prior to crystallization, the frozen protein samples were thawed and supplemented with the appropriate concentration of mupirocin (Sigma-Aldrich, cat: 07188) (PmIleRS2 and mutants: 10 mM final concentration, PmIleRS1 and mutants: 3 mM final concentration), or Ile-AMS (synthetized in-house, Supplementary Fig. 15) (2.5 mM final concentration). The solutions were centrifuged for 10 min at 4 °C and 25,000 × *g* to remove particulates. The supernatants were used for sitting drop vapor diffusion crystallization experiments in 24-well Cryschem S Plates (Hampton, cat: HR3-159) containing 300 µl of well solution as precipitant. The drops were set up by mixing 1 µl of protein:ligand solution with 1 µl of the well-solution, and crystals were obtained by incubation for one week at 4 °C, except for wt-PmIleRS1:mupirocin, which crystallized at 19 °C. Single crystals of wt-PmIleRS1:Ile-AMS or mut-GMHH-PmIleRS1:Ile-AMS complexes grew in crystallization buffer containing 300 mM Li₂SO₄ (Sigma Aldrich, cat: L6375), 150 mM Na₂SO₄ (Sigma Aldrich, cat: S6547), and 15−25% PEG3350 (Sigma-Aldrich, cat: P4338). Crystal clusters of the wt-PmIleRS1:mupirocin complex were obtained from a crystallization buffer with 400−900 mM LiCl (Sigma Aldrich, cat: L4408) and 10−20% PEG3350 (Sigma-Aldrich, cat: P4338). wt-PmIleRS2:Ile-AMS, wt-PmIleRS2:mupirocin, mut-GVHH-PmIleRS2:Ile-AMS and W130Q-PmIleRS2:Ile-AMS crystallized in a buffer containing 0.2−0.5 M ammonium tartrate (Sigma-Aldrich, A2956) and 10−20% PEG3350 (Sigma-Aldrich, cat: P4338). For wt-HMGH-PmIleRS1:Ile-AMS and mut-GMHH-PmIleRS1:Ile-AMS crystals, 20% glycerol (Sigma-Aldrich, cat: G6279) (final concentration) was used as cryo-protectant, while for wt-HMGH-PmIleRS1:mupirocin, wt-HVGH-PmIleRS2:Ile-AMS, wt-HVGH-PmIleRS2:mupirocin, mut-GVHH-PmIleRS2:Ile-AMS and W130Q-PmIleRS2:Ile-AMS, crystals were stabilized and cryo-protected using PEG400 (Sigma-Aldrich, cat: 202398) at a final concentration of 20%. Prior to data collection, all crystals were mounted on nylon loops (Hampton) and flash-frozen in liquid nitrogen.

## X-ray data collection and structural analysis
Diffraction data of cryo-protected crystals were collected at 100 K using a wavelength of 1 Å at Beamline X06SA (PXI) or X06DA (PXIII) at the Paul Scherrer Institut (PSI) Villigen (Switzerland) and integrated and scaled using the XDS package (SV: 20190806)[44].

Initial phases for the wt-PmIleRS1:Ile-AMS complex were obtained by molecular replacement (CCP4i PHASER module, SV: 7.0.077) using the main body of SaIleRS1:tRNA:mupirocin (1QU2) and the TtVaIRS editing domain (1WKA) as search models. The structures of the mut-GMHH-PmIleRS1:Ile-AMS and wt-HMGH-PmIleRS1:mupirocin complexes were determined by molecular replacement using our solved wt-HMGH-PmIleRS1:Ile-AMS structure. Molecular replacement phases for the wt-HVGH-PmIleRS2:mupirocin dataset were obtained using the wt-HVGH-TtIleRS2:mupirocin structure (1JZS) as search model, and the structure was completed utilizing the *Phenix.autobuild* module in PHENIX (SV: 1.20.1_4487)[45] and manual model building in Coot (SV: 0.8.9.2)[46]. Initial phases for wt-HVGH-PmIleRS2:Ile-AMS, mut-GVHH-PmIleRS2:Ile-AMS, W130Q-PmIleRS2:Ile-AMS were obtained by molecular replacement using our wt-HVGH-PmIleRS2:mupirocin coordinates as search model.

All structures were finalized using iterative rounds of manual model building with Coot followed by coordinate and individual B-factor refinement in Phenix. To interpret the poorly resolved parts of the C-terminal tRNA-binding domain of both wt-HMGH-PmIleRS1 and mut-GMHH-PmIleRS1, we first docked an AlphaFold2 (SV: 2.3.2)[47] model of this domain as a rigid body, manually adjusted the connections to the neighboring domains and refined the completed model as described above. Refinement and validation statistics for the final models are listed in Supplementary Table 2.

Inspection of our 1.9 Å mupirocin-bound PmIleRS2 structure clearly indicated an inverted chirality in the monic acid moiety of mupirocin. To account for the discrepancy between our ligand structure and the publicly available coordinate and topology files of the mupirocin (MRC) ligand, the chirality was manually changed in the restraints.

### Protein sequence retrieval
Sequences of the IleRS domains were retrieved using Pfam HMM profile (PF00133) from a set of 738 representative archaeal and bacterial genomes[24] that span the prokaryote tree of life using HMMsearch (SV: 2.41.2) with the Pfam-defined gathering threshold[48]. Incomplete sequences, i.e., those lacking the catalytic motif, were removed. Next, to remove redundancy, the remaining sequences were further clustered at 70% identity with CD-HIT (SV: 4.8.1)[49]. Additionally, four sequences of ValRS—two bacterial and two archaeal—were selected from the species set and were used as outgroup. Overall, 374 sequences were taken and aligned using MAFFT (SV: 7.505) with the *–linsi* option[50]. The alignment was then trimmed with trimAl (SV: 1.2) to remove gap-rich positions (*--gappyout* option)[51].

### Phylogenetics and ancestral sequence reconstruction
The resulting alignment was then used to build the phylogenetic tree of the family with the FastTree software (SV: 2.1) with Jones−Taylor−Thornton (JTT) evolutionary models and the following parameters: *-pseudo, -spr 4, -mlacc 2, -slownni*[52]. The resulting tree was rooted with the ValRS outgroup. Ancestral sequence reconstruction was performed using *codeml* from the PAML-X package (SV: 1.3.1)[53] with the empirical JTT model and default parameters. Resulting trees were plotted and annotated using the program package ITOL (SV: 6.7)[54]. The resulting tree, annotated with ancestral nodes, shown in Supplementary Fig 4, is provided as a Source data file.

### Molecular dynamics simulations
Four starting structures for molecular-dynamics (MD) simulations were used. In addition to the crystal structures of IleRS2 complexed to Ile-AMS or mupirocin, two apo-IleRS2 structures were generated by deleting the ligands. The parameters for mupirocin and Ile-AMS were prepared de novo (Supplementary Fig. 13), whereas the IleRS2 enzyme was described using the Amber ff14SB force field[55]. All models were dissolved in an explicit TIP3P water model[56] and placed in a truncated octahedron-shaped simulation box. The systems were minimized in

several steps and equilibrated at a temperature of 27 °C for 5 ns, followed by the production runs at the same temperature. Each model was simulated for 360 ns, and each trajectory consisted of 180,000 structures. The SHAKE algorithm was applied[57], and a 2-fs step was used for numerical integration. The temperature was maintained using Langevin dynamics, and the pressure was controlled by Berendsen barostat[58]. The cut-off value was set to 9 Å. Production runs were performed with the AMBER20 (SV: 20.0) on GPU using the pmemd.CUDA engine[59]. A complete description of the MD simulation system setup is given in Supplementary Table 4.

### Molecular dynamics data analysis
The analysis was performed using the cpptraj module[60] within the AmberTools20 package (SV: 20.0). The evolution of the secondary structure elements during the simulation time for helix α2 was monitored. Elements of the secondary structure were assigned according to the DSSP classification[61] based on the analysis of φ and ψ torsion angles and hydrogen bonds.

### Synthesis of 5′-O-[(L-isoleucyl)sulfamoyl]adenosine (Ile-AMS)
Sulfamoyl chloride and compounds 1–3 (Supplementary Fig. 15) were prepared according to refs. 62,63 with some changes detailed in Supplementary Methods. Reagents and solvents for the synthesis of the compounds were obtained from Sigma-Aldrich Corp. (Germany) and Bachem (Switzerland). Organic solvents were further purified and/or dried using standard methods. Thin layer chromatography (TLC) was performed on 0.25 mm TLC Silica gel 60 $F_{254}$ plates (Merck, cat: 105554). Visualization was achieved using UV light at 254 nm and ninhydrine. Column chromatography was performed on Silica gel 60 (size 70–230 mesh ASTM) column material (Millipore, cat: 107734). The ATR FT-IR spectrum was recorded on a FT-IR Perkin-Elmer Spectrum Two device (4000–400 $cm^{-1}$ region). Mass spectra (ESI-MS) were recorded on an Agilent Technologies 1200 series HPLC system. $^{1}H$ and $^{13}C$ NMR spectra of all precursors were recorded on a AV-III HD Bruker spectrometer at 400 MHz ($^{1}H$) and 100 MHz ($^{13}C$). All NMR experiments were performed at 298 K. Chemical shifts were referenced with respect to tetramethylsilane. The details about the MS and NMR protocols are given in Supplementary Methods.

## Data availability
The structural data generated in this study were deposited in the Protein Data Bank (PDB) under accession numbers: 8C8U (wt-HVGH-PmIleRS2:mupirocin), 8C8V (wt-HVGH-PmIleRS2:Ile-AMS), 8C8W (mut-GVHH-PmIleRS2:Ile-AMS), 8C9D (W130Q-PmIleRS2:Ile-AMS), 8C9E (wt-HMGH-PmIleRS1:Ile-AMS), 8C9F (mut-GMHH-PmIleRS1:Ile-AMS) and 8C9G (wt-HMGH-PmIleRS1:mupirocin). The structural data used in this study are available in PDB under accession codes 1QU2, 1WKA, 6LDK, 7D5C, and 1JZS. Source data generated in this study are provided in the Source data file, which contains phylogenetic data (Fig. 2 and Supplementary Fig. 4), kinetic data (Fig. 1 and Supplementary Fig. 2), de novo parameters used in MD simulations (Supplementary Fig. 13) and IR, NMR, and MS spectra (Supplementary Fig. 15). Starting molecular structures, topologies, and parameter files for MD simulations are deposited on a publicly available GitHub repository accessible through the following link: https://github.com/aleksandra-mar/DATA_SHARING/releases/latest. Other relevant data are either contained in the manuscript or provided in the Supplementary Information file. Source data are provided with this paper.

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

## Acknowledgements

This work was supported by the SNSF, Swiss Enlargement Contribution in the framework of the Croatian-Swiss Research Programme, Grant IZHRZO_180567 (to I.G.-S and N.B.) and European Regional Development Fund (infrastructural project CIuK, grant number KK.01.1.1.02.0016). A.M. acknowledges the Zagreb University Computing Centre (SRCE) for granting computational resources on the ISABELLA cluster. We would like to thank Jeff Errington for careful reading of the manuscript and Dan S. Tawfik, Dragana Despotovic, and Liam M. Longo for numerous fruitful discussions.

## Author contributions
I.G.-S. and A.B. conceptualized and planned the experiments. A.B. cloned and purified the enzymes, performed kinetic and structural analyses, and prepared all figures. M.L. solved the crystal structure of the wt-HVGH-PmIleRS2:mupirocin complex and supervised the crystallographic work of A.B. J.J. performed the phylogenetic analysis and ancestral sequence reconstruction. V.Z. cloned mut-GMHH-BmIleRS1 and kinetically characterized the mut-GVHH-PmIleRS2 enzyme. V.P.-P. and Z.C. synthesized the Ile-AMS analog. A.M. performed molecular dynamics simulations. I.G.-S., A.B., M.L., and N.B analyzed the data. I.G.-S. and N.B. conceived and supervised the project. I.G.-S. wrote the manuscript with the contribution from all authors.

## Competing interests
The authors declare no competing interests.
