## [Peer Review File · Nature Communications]

Antibiotic hyper-resistance in a class I aminoacyl-tRNA synthetase with altered active site signature motifREVIEWER COMMENTS

Reviewer #1 (Remarks to the Author):

Mupiricin is the best known antibiotic against ARSs (specifically IleRS), and is an effective topical therapy against *S. aureus*. Its effectiveness is blunted by the emergent of resistance, which is known to be associated with the transfer and spread of an IleRS2 form that is likely eukaryotic in origin. While the structure of an *S. aureus* IleRS – tRNA-mupiricin complex and a *T. thermophilus* IleRS – mupiricin complex were determined years ago, details of the basis of resistance have remained obscure, and the mechanism has remained unknown. Here, Brkic et al apply bioinformatics, mutagenesis studies, and X-ray crystallography to the characterization of IleRS1 and IleRS2 variants to investigate the basis of mupiricin resistance and hyper resistance. Their overall result, which is well supported by the presented data, is that the basis of mupiricin resistance is linked to sequence and structural alterations of the Class I HIGH signature sequence motif. While conventional wisdom might hold that mutation of one of the key conserved motifs might be fatal to the enzyme's activity, these data illustrate how this changes in the two Class I signature sequences can be accommodated. Owing to the general insights this story provides into how enzymes achieve resistance to clinically significant antibiotics, this work is of important general interest.

Major Findings of this work

1. Non-canonical HIGH sequences are rare in IleRS, but limited only to IleRS2
2. The introduction of a non-canonical HIGH signature sequence into IleRS2 confers increased resistance to Mupiricin. These sequences don't significantly decrease catalytic function.
3. By contrast, when GVHH is inserted into IleRS1, there is a significant loss of catalytic function, and no resistance to Mupiricin.
4. crystal structure of IleRS1 and IleRS2 from *P. megaterium* comparing both Ile-AMS and mupiricin were determined.
5. Authors identified the structural basis of mupiricin resistance, linking it to conformational changes of the HIGH motif and its interactions with the KMSKS loop. The basis of Hyper-resistance inferred via a modeling exercise.
6. The data collectively assess IleRS structural features associated with binding Ile-AMS and Mupiricin.

Overall Review

The central argument of the paper is that mupiricin resistance occurs when residues in the active site are discouraged from forming a set of highly specific interactions with mupiricin, including a salt bridge with the 9-hydroxynonanoic moiety and hydrogen bonds to the oxygen atoms in the monic acid moiety. The resistant forms of the enzyme don't form these interactions because the HIGH motif is displaced towards the KMSKS loop. This mispositioning occurs more readily when HIGH motif undergoes substitution. The data presented by the authors – especially the crystallographic- strongly support this hypothesis and the general conclusion that resistance emerges principally from changes in active site structure in close proximity to the antibiotic. Technically, there are no concerns with the data or its presentation. The bioinformatics, enzymology, and crystallography are all well executed with plenty supplementary support to indicate high data quality.

An area that the paper does not explore too deeply is the connection between catalytic function and drug resistance. Given that a common set of residues participate in both functions, this is a critical issue. The data show that the IleRS2 enzyme readily accommodates both the canonical HIGH and non-canonical GVHH motifs without losing catalytic activity; the former retains sensitivity to mupiricin. By contrast, when the non-canonical motif is swapped into the IleRS1 enzyme, it neither gains resistance nor retains catalytic activity. This lack of reciprocity suggests that there is still more to understand about the relationship between catalytic function and mupiricin resistance (see Note 1). While the authors have provided evidence that rearrangement of helix $\alpha 1$ constitutes a major element in the permissiveness of IleRS2, there might be additional global structural features which remain to be identified (see Note 2). These observations raise the question of how the active site change that promotes Mup resistance impacts on catalytic

mechanism. Does IleRS2 employ different reaction mechanisms than IleRS1s, or rely on various catalytic residues to different extents?

There are some additional questions that bear some deeper discussion. Notably, the analysis and the structures determined in the absence of tRNA. This represents a caveat to the conclusions that the authors should consider. While we know that IleRS is not an ARS that requires tRNA for the first step, it might be worth asking whether or not the structural changes seen in the comparison of the Ile-AMS and mupirocin structure might be altered by the presence of tRNA (see Note 3 below). (It is appreciated that this might be speculative.) This question is linked to the relative conformational mobility of the IleRS1 and IleRS2 forms (Note 4). Crystallography can provide a "snapshot" of the structure(s) but dynamic information is lacking; this might have to be interrogated by molecular dynamics, as the authors report Supp Figure 12. A third question concerns the phenomenon of "hyper-resistance". As yet, the authors base their conclusions on this point on modeling experiments; query whether actual experimental support is needed.

Notes

1. Page 10, line 159: Authors claim GVHH substitution into both *T. thermophilus* and *P. Meg* IleRS2 enzymes did not "strongly" affect catalytic function. Inspection of Table 1 shows that this is more true of *T. thermophilus* than *P. meg*. For the former, there was 5X decrease in *k_{cat}* accompany the 150X increase Mup Ki. For the latter, the 460X increase in Mup Ki was accompanied by a 10X decrease in *K_{cat}*. So, there is some kind of trade off.
2. During their bioinformatics analysis, did the authors look for any co-varying residues in IleRS that accompany the inheritance of a non-canonical HIGH (e.g. GVHH) motif. Such residues
3. Authors do not comment on the possibility that the presence of the tRNA might well affect these parameters. May not alter general conclusions, but might alter the specifics
4. Page 12, structural comparison: what does the crystallography imply about the mobility of these various regions?
5. On page 22, the authors write, "in bacteria two IleRS types, IleRS1 and IleRS2 alternatively perform the housekeeping function." How strongly has this been established? In the absence of Mup selection, can bacteria tolerate IleRS2 deletion without a loss of fitness?

Reviewer #2 (Remarks to the Author):

This manuscript describes a study that is both a highly relevant medical result and fascinating exploration of how aminoacyl-tRNA synthetase (aaRS) mechanisms are interwoven into the evolution of tertiary structural plasticity. Pseudomonic acid is one of a very small number—perhaps the only—of bacteriospecific aaRS inhibitors that have proven clinically useful in controlling bacterial infections. That unique success is surprising because of multiple lines of evidence that many of the synthetases have sharply divided phylogenetic trees with distinct eukaryotic/archaeal clades. Moreover, the existence of multiple naturally occurring compounds with quite highly selective inhibition of bacterial forms suggests that Nature has learned to exploit those differences in waging their internecine chemical warfare.

Not surprisingly, bacteria have also evolved defenses against inhibition of isoleucyl-tRNA synthetase (IleRS) by pseudomonic acid. The authors use an elegant combination of phylogenetic, structural, and biochemical evidence to examine the molecular mechanisms of mupirocin resistance. Their results are both subtle and interwoven with unexplored dimensions of the mechanistic plasticity of protein active sites. IleRS is a member of the Class I superfamily, which is defined by the use of two catalytic signatures, HXGH (X= large hydrophobe) and KMSKS, in transition-state stabilization. Although a crystal structure of a mupirocin-bound form of *S. aureus* IleRS was described decades ago, that structure in isolation brought little insight either into the detailed mechanism of mupirocin inhibition or to how Nature circumvented it in resistant forms of the enzyme.

More recently it emerged that resistant forms of the enzyme use a bizarre rearrangement of the

two histidines in the HXGH signature in which the glycine and first histidine exchange positions, to form the altered signature, GVHH. The central strategy in this paper is construction of variant forms of the sensitive (IleRS1) and resistant (IleRS2) in which the two signatures are investigated in both enzymes. Analysis of these hybrid constructs opens windows on how mupirocin binds so tightly; how the altered GVHH signature enables the resistant, but not the sensitive form to both catalyze both amino acid activation and aminoacylation while avoiding inhibition by mupirocin; and provides an entirely new window on how the plasticity of the Rossmann nucleotide binding domain enables the implicit molecular sleight of hand.

The experiments show:

1. The GVHH signature is non-functional in the IleRS1 background.
2. Why the GVHH signature is functional in the IleRS2 background.
3. How the GVHH signature endows IleRS2 with mupirocin resistance.
4. How structural changes necessary to allow the GVHH signature to become functional also enable construction of a hyper-resistant IleRS2 based on a resistant IleRS that contains the canonical, instead of the GVHH signature, confirming the origin of the hyper-resistance of naturally occurring IleRS2 enzymes that contain the non-canonical signature.

Added interest attaches to the structural changes that enable mupirocin resistance in IleRSs that contain a canonical HXGH signature, because they are not permitted in the IleRS1 background. The ensemble of structures presented here thus provide fertile ground for next level structural analysis, opening new windows by posing the question of why those changes are permitted in one variant of the Rossmann fold but not the other.

The manuscript is logically constructed, well-written, and easy to read. The experiments are all sound. It is eminently suitable for publication in Nature Communications.

There is only one suggestion to the authors, which is to amplify/clarify the discussion at the top of page 5. For clarity and biological insight, it would be useful in this section to provide information about the organism that makes pseudomonic acid (mupirocin) with particular reference to the organisms from which the crystal structures were derived. It is confusing to sort out the differences between the IleRS2 versions with a canonical and those with a non-canonical HXGH signature.

Reviewer #3 (Remarks to the Author):

I find this paper to be well-written and thoroughly researched. The authors have demonstrated a strong understanding of the relevant literature, and their methodology is sound. The results are significant and the conclusions are well-supported by the data presented. Overall, I would recommend publishing this paper in NC.

However, there are still some minor issues that need to be resolved before publication. I recommend replication experiments be conducted as the results of single molecule dynamics simulations can be misleading, the general rule is that at least three replicates should be performed to verify the reliability and stability of simulation results.

Reviewer #1 (Remarks to the Author):

Mupiricin is the best known antibiotic against ARSs (specifically IleRS), and is an effective topical therapy against *S. aureus*. Its effectiveness is blunted by the emergent of resistance, which is known to be associated with the transfer and spread of an IleRS2 form that is likely eukaryotic in origin. While the structure of an *S. aureus* IleRS:tRNA-mupiricin complex and a *T. thermophilus* IleRS:mupiricin complex were determined years ago, details of the basis of resistance have remained obscure, and the mechanism has remained unknown. Here, Brkic et al apply bioinformatics, mutagenesis studies, and X-ray crystallography to the characterization of IleRS1 and IleRS2 variants to investigate the basis of mupiricin resistance and hyper resistance. Their overall result, which is well supported by the presented data, is that the basis of mupiricin resistance is linked to sequence and structural alterations of the Class I HIGH signature sequence motif. While conventional wisdom might hold that mutation of one of the key conserved motifs might be fatal to the enzyme's activity, these data illustrate how this changes in the two Class I signature sequences can be accommodated. Owing to the general insights this story provides into how enzymes achieve resistance to clinically significant antibiotics, this work is of important general interest.

Major Findings of this work

1. Non-canonical HIGH sequences are rare in IleRS, but limited only to IleRS2
2. The introduction of a non-canonical HIGH signature sequence into IleRS2 confers increased resistance to Mupiricin. These sequences don't significantly decrease catalytic function.
3. By contrast, when GVHH is inserted into IleRS1, there is a significant loss of catalytic function, and no resistance to Mupiricin.
4. crystal structure of IleRS1 and IleRS2 from *P. megaterium* comparing both Ile-AMS and mupiricin were determined.
5. Authors identified the structural basis of mupiricin resistance, linking it to conformational changes of the HIGH motif and its interactions with the KMSKS loop. The basis of Hyper-resistance inferred via a modeling exercise.
6. The data collectively assess IleRS structural features associated with binding Ile-AMS and Mupiricin.

Overall Review

The central argument of the paper is that mupiricin resistance occurs when residues in the active site are discouraged from forming a set of highly specific interactions with mupiricin, including a salt bridge with the 9-hydroxynonanoic moiety and hydrogen bonds to the oxygen atoms in the monic acid moiety. The resistant forms of the enzyme don't form these interactions because the HIGH motif is displaced towards the KMSKS loop. This mispositioning occurs more readily when HIGH motif undergoes substitution. The data presented by the authors - especially the crystallographic- strongly support this hypothesis and the general conclusion that resistance emerges principally from changes in active site structure in close proximity to the antibiotic. Technically, there are no concerns with the data or its presentation. The bioinformatics, enzymology, and crystallography are all well executed with plenty supplementary support to indicate high data quality.

An area that the paper does not explore too deeply is the connection between catalytic function and drug resistance. Given that a common set of residues participate in both functions, this is a critical issue. The data show that the IleRS2 enzyme readily accommodates both the canonical HIGH and non-canonical GVHH motifs without losing catalytic activity; the former retains sensitivity to mupiricin. By contrast, when the non-canonical motif is swapped into the IleRS1 enzyme, it neither gains resistance nor retains catalytic activity. This lack of reciprocity suggests that there is still more to understand about the relationship between catalytic function and mupiricin resistance (see Note 1). While the authors have provided evidence that rearrangement of helix a1 constitutes a major element in the permissiveness of IleRS2, there might be additional global structural features which remain to be identified (see Note 2). These

observations raise the question of how the active site change that promotes Mup resistance impacts on catalytic mechanism. Does IleRS2 employ different reaction mechanisms than IleRS1s, or rely on various catalytic residues to different extents?

There are some additional questions that bear some deeper discussion. Notably, the analysis and the structures determined in the absence of tRNA. This represents a caveat to the conclusions that the authors should consider. While we know that IleRS is not an ARS that requires tRNA for the first step, it might be worth asking whether or not the structural changes seen in the comparison of the Ile-AMS and mupiricin structure might be altered by the presence of tRNA (see Note 3 below). (It is appreciated that this might be speculative.) This question is linked to the relative conformational mobility of the IleRS1 and IleRS2 forms (Note 4). Crystallography can provide a “snapshot” of the structure(s) but dynamic information is lacking; this might have to be interrogated by molecular dynamics, as the authors report Supp Figure 12. A third question concerns the phenomenon of “hyper-resistance”. As yet, the authors base their conclusions on this point on modeling experiments; query whether actual experimental support is needed.

Notes

1. Page 10, line 159: Authors claim GVHH substitution into both *T. thermophilus* and *P. Meg* IleRS2 enzymes did not “strongly” affect catalytic function. Inspection of Table 1 shows that this is more true of *T. thermophilus* than *P. meg*. For the former, there was 5X decrease in *k_{cat}* accompany the 150X increase Mup *K_i*. For the latter, the 460X increase in Mup *K_i* was accompanied by a 10X decrease in *K_{cat}*. So, there is some kind of trade off.
2. During their bioinformatics analysis, did the authors look for any co-varying residues in IleRS that accompany the inheritance of a non-canonical HIGH (e.g. GVHH) motif. Such residues
3. Authors do not comment on the possibility that the presence of the tRNA might well affect these parameters. May not alter general conclusions, but might alter the specifics
4. Page 12, structural comparison: what does the crystallography imply about the mobility of these various regions?
5. On page 22, the authors write, “in bacteria two IleRS types, IleRS1 and IleRS2 alternatively perform the housekeeping function”. How strongly has this been established? In the absence of Mup selection, can bacteria tolerate IleRS2 deletion without a loss of fitness?

Reviewer #2 (Remarks to the Author):

This manuscript describes a study that is both a highly relevant medical result and fascinating exploration of how aminoacyl-tRNA synthetase (aaRS) mechanisms are interwoven into the evolution of tertiary structural plasticity. Pseudomonic acid is one of a very small number, perhaps the only, of bacteriospecific aaRS inhibitors that have proven clinically useful in controlling bacterial infections. That unique success is surprising because of multiple lines of evidence that many of the synthetases have sharply divided phylogenetic trees with distinct eukaryotic/archaeal clades. Moreover, the existence of multiple naturally occurring compounds with quite highly selective inhibition of bacterial forms suggests that Nature has learned to exploit those differences in waging their internecine chemical warfare.

Not surprisingly, bacteria have also evolved defenses against inhibition of isoleucyl-tRNA synthetase (IleRS) by pseudomonic acid. The authors use an elegant combination of phylogenetic, structural, and biochemical evidence to examine the molecular mechanisms of mupirocin resistance. Their results are both subtle and interwoven with unexplored dimensions of the mechanistic plasticity of protein active sites. IleRS is a member of the Class I superfamily, which is defined by the use of two catalytic signatures, HXGH (X= large hydrophobe) and KMSKS, in transition-state stabilization. Although a crystal structure of a

mupirocin-bound form of *S. aureus* IleRS was described decades ago, that structure in isolation brought little insight either into the detailed mechanism of mupirocin inhibition or to how Nature circumvented it in resistant forms of the enzyme.

More recently it emerged that resistant forms of the enzyme use a bizarre rearrangement of the two histidines in the HXGH signature in which the glycine and first histidine exchange positions, to form the altered signature, GVHH. The central strategy in this paper is construction of variant forms of the sensitive (IleRS1) and resistant (IleRS2) in which the two signatures are investigated in both enzymes. Analysis of these hybrid constructs opens windows on how mupirocin binds so tightly; how the altered GVHH signature enables the resistant, but not the sensitive form to both catalyze both amino acid activation and aminoacylation while avoiding inhibition by mupirocin; and provides an entirely new window on how the plasticity of the Rossmann nucleotide binding domain enables the implicit molecular sleight of hand.

The experiments show:

1. The GVHH signature is non-functional in the IleRS1 background.
2. Why the GVHH signature is functional in the IleRS2 background.
3. How the GVHH signature endows IleRS2 with mupirocin resistance.
4. How structural changes necessary to allow the GVHH signature to become functional also enable construction of a hyper-resistant IleRS2 based on a resistant IleRS that contains the canonical, instead of the GVHH signature, confirming the origin of the hyper-resistance of naturally occurring IleRS2 enzymes that contain the non-canonical signature.

Added interest attaches to the structural changes that enable mupirocin resistance in IleRSs that contain a canonical HXGH signature, because they are not permitted in the IleRS1 background. The ensemble of structures presented here thus provide fertile ground for next level structural analysis, opening new windows by posing the question of why those changes are permitted in one variant of the Rossmann fold but not the other.

The manuscript is logically constructed, well-written, and easy to read. The experiments are all sound. It is eminently suitable for publication in Nature Communications.

There is only one suggestion to the authors, which is to amplify/clarify the discussion at the top of page 5. For clarity and biological insight, it would be useful in this section to provide information about the organism that makes pseudomonic acid (mupirocin) with particular reference to the organisms from which the crystal structures were derived. It is confusing to sort out the differences between the IleRS2 versions with a canonical and those with a non-canonical HXGH signature.

Reviewer #3 (Remarks to the Author):

I find this paper to be well-written and thoroughly researched. The authors have demonstrated a strong understanding of the relevant literature, and their methodology is sound. The results are significant and the conclusions are well-supported by the data presented. Overall, I would recommend publishing this paper in NC.

However, there are still some minor issues that need to be resolved before publication. I recommend replication experiments be conducted as the results of single molecule dynamics simulations can be misleading, the general rule is that at least three replicates should be performed to verify the reliability and stability of simulation results.

RESPONSES TO REVIEWERS

First, we would like to thank the reviewers on their comprehensive and careful reviews that improve quality of the manuscript.

REVIEWER 1:

The reviewer: “The data show that the IleRS2 enzyme readily accommodates both the canonical HIGH and non-canonical GVHH motifs without losing catalytic activity; the former retains sensitivity to mupiricin. By contrast, when the non-canonical motif is swapped into the IleRS1 enzyme, it neither gains resistance nor retains catalytic activity. This lack of reciprocity suggests that there is still more to understand about the relationship between catalytic function and mupiricin resistance (see Note 1)

Note 1. Page 10, line 159: Authors claim GVHH substitution into both *T. thermophilus* and *P. meg* IleRS2 enzymes did not “strongly” affect catalytic function. Inspection of Table 1 shows that this is more true of *T. thermophilus* than *P. meg*. For the former, there was 5X decrease in k_{cat} accompany the 150X increase Mup K_i . For the latter, the 460X increase in Mup K_i was accompanied by a 10X decrease in K_{cat} . So, there is some kind of trade off.”

Response: *We agree with the reviewer that introducing GVHH in TlleRS2 and PmIleRS2 brings hyper-resistance at a trade-off with the catalytic activity. We report on the 10-fold effect as “not strong” because we expected a higher effect based on mutation of the key catalytic motif. We also agree that the effect is lower for TtleRS2. This relates to our finding that a natural reversion of the non-canonical (GVHH) to the canonical (HVGH) motif took place in the Deinococcus-Thermus clade (Fig 2). That said, TtleRS2, which has the canonical motif but descends from the ancestor with the non-canonical motif, likely has the active site better (pre)adjusted for the GVHH introduction.*

To address the reviewer’s point, we changed the sentence “Introducing the non-canonical GVHH motif into PmIleRS2 and TtleRS2 also did not strongly affect the enzymes – the variants showed increased K_M and decreased k_{cat} values of up to 10-fold” into

Introducing the non-canonical GVHH motif into PmIleRS2 and TtleRS2 also did not strongly affect the enzymes, yet hyper-resistance, in this case, came at the expense of increased K_M and decreased k_{cat} values of up to 10-fold. (Page 6, lines 137-140)

and introduced a novel sentence in the revised version (Page 7, lines 161-162)

That said, laboratory exchange of HVGH back to GVHH in TtleRS came at a minor expense of its catalytic efficiency (Table 1).

The reviewer: While the authors have provided evidence that rearrangement of helix a1 constitutes a major element in the permissiveness of IleRS2, there might be additional global structural features which remain to be identified (see Note 2). These observations raise the question of how the active site change that promotes Mup resistance impacts on catalytic mechanism. Does IleRS2 employ different reaction mechanisms than IleRS1s, or rely on various catalytic residues to different extents?

Note 2. During their bioinformatics analysis, did the authors look for any co-varying residues in IleRS that accompany the inheritance of a non-canonical HIGH (e.g. GVHH) motif.

Response: *We did not observe evidence for different catalytic mechanisms between IleRS1 and IleRS2. PmIleRS1 and PmIleRS2 bind the analogue of the reaction intermediate highly similarly (Supplementary Fig. 7), which supports utilisation of the*

same catalytic mechanisms. Further, mutation of the natural ALHH motif of DrIleRS2 to ALGH led to the loss of activity, showing that IleRS2 enzymes also require the 1st/3rd His for the activity (i.e. implying the same reaction mechanisms).

To address this point, we changed the sentence “This allows accommodation of the 3rd His in mut-GVHH-PmIleRS2 in a productive manner without a clash with the adenine moiety (Fig. 4B).” into

This allows accommodation of the 3rd His in mut-GVHH-PmIleRS2 without displacement of the adenine moiety. This way, the 3rd His can take over the role of the 1st His in the canonical reaction mechanism (Fig. 4B). (Page 9, lines 216-218)

We agree with the reviewer that the active sites of IleRS1 and IleRS2 differ in structural features allowing rearrangement of the α 1-helix solely in IleRS2. We were eager to understand the basis of it and mutated several conserved positions, within and around helix 1, in IleRS1 by introducing the analogous residues found in IleRS2. The same was done for IleRS2. Any mutation introduced in IleRS1 abolished the enzyme catalytic activity (Table 1. of this response). In contrast, the IleRS2 variants retained their activity upon reciprocal residue exchange, however, no IleRS1-specific features, such as lower K_i for mupirocin, were found in these IleRS2 variants (Table 1 of this response and Supplementary Fig. 8). Thus, the structural changes underlying the IleRS1 and IleRS2 divisions are likely more global and rooted in the distinct architecture of the two IleRS active sites.

Table 1. Catalytic activity of the IleRS1 and IleRS2 mutants with exchanged reciprocal residues measured in the activation step

PmIleRS1	
Q136W	no activity
I63P/N70G	no activity
I63P/N70G/Q136W	no activity
I63P/N70G GMHH	no activity
I63P/N70G/Q136W GMHH	no activity
N70G/I63P/F140A/Q136W	10 ⁴ -fold drop in the rate
PmIleRS2	
W130Q/Y126S	$K_M(\text{L-Ile}) = 18.7 \pm 1.3 \mu\text{M}$, $K_M(\text{ATP}) = (2.8 \pm 0.2) \times 10^3 \mu\text{M}$, $k_{\text{cat}} = 5.25 \pm 0.07 \text{ s}^{-1}$, $K_i(\text{Mup}) = 3.2 \pm 0.1 \mu\text{M}$
P53A/G60N/R61K	$K_M(\text{L-Ile}) = 191 \pm 10 \mu\text{M}$, $K_M(\text{ATP}) = (2.6 \pm 0.1) \times 10^3 \mu\text{M}$, $k_{\text{cat}} = 6.06 \pm 0.07 \text{ s}^{-1}$, $K_i(\text{Mup}) = 3.1 \pm 0.2 \mu\text{M}$
P53A/G60N/R61K/W130Q/Y126S	$K_M(\text{L-Ile}) = 68 \pm 2 \mu\text{M}$, $K_M(\text{ATP}) = (7.5 \pm 0.2) \times 10^3 \mu\text{M}$, $k_{\text{cat}} = 17.0 \pm 0.2 \text{ s}^{-1}$, $K_i(\text{Mup}) = 21.1 \pm 0.7 \mu\text{M}$

Following the reviewer’s suggestion we used the bioinformatics tool SDPpred (<http://bioinf.fbb.msu.ru/SDPpred/>) to search for the residues differently conserved in IleRS1 and IleRS2. This approach revealed around 40 positions within the catalytic and the CP domains (Pfam HMM profile PF00133, Supplementary Dataset 1) that are differently conserved, at a significant level, between the two bacterial IleRS types. These residues are given in Table 2 of this response and visualised in the accompanying PyMOL session (SDPpred.pse) by orange and blue sticks in the PmIleRS1 and PmIleRS2 structures, respectively. The residues that were mutated and kinetically tested (Table 1 above, Supplementary Fig. 8) are highlighted by thicker sticks. We also enclose an alignment file used for SDPpred analysis. The differently conserved residues are distributed throughout the CP and catalytic domains (Ile-AMS binding pocket generally excluded), supporting our conclusion that the changes responsible for the different rearrangements of the α 1-helix, which enables permissiveness for the non-canonical motif solely in IleRS2, are deeply rooted in the distinct architecture of the two IleRS active site folds.

In the initial version of the manuscript we stated: “Taken together, our results support a view that the differences between IleRS1 and IleRS2 are rooted deeply in the overall architecture of the catalytic domain, which is indicative for separate evolution trajectories of the IleRS1 and IleRS2 catalytic folds, likely because of distinct selection forces.” This sentence summarizes our understanding of the system best and is kept in the revised version without modifications (Page 16, lines 376-379). SDPpred analysis also explains why reengineering IleRS1 into IleRS2 and vice versa was/is highly challenging. In our opinion introducing the mutational analysis or SDPpred analysis in the revised version will not improve the manuscript and therefore we provided it for reviewing purposes only.

Table 2. Residues differently conserved in IleRS1 and IleRS2 identified using SDPpred. The difference is significant when the parameter mutual information is above 0.5

Alignment position	Position in PmIleRS1	Distribution IleRS1	Position in PmIleRS2	Distribution IleRS2	Mutual information
26	52L	L (97.5%)	42F	F (94.9%)	0.58
27	53H	H (97.5%)	43Y	Y (81.1%)	0.67
32	58Y	Y (100 %)	48T	T (56.6%), F (30.3%)	0.67
37	63I	I (71.5%), L (24.1%)	53P	P (93.1%)	0.67
44	70N	N (100%)	60G	A (45.1%), T (30.3%)	0.67
45	71K	K (98.7%)	61R	R (69.7%), G (25.1%)	0.66
64	90Y	Y (87.3%)	80R	R (92%)	0.55
65	91V	V (75.9%)	81K	K (53.7%), R (36%)	0.58
66	92P	P (96.2%)	82A	A (46.9%), F (28.6%)	0.65
94	122R	R (96.8%)	116I	N (80%)	0.65
101	129A	A (100%)	123V	V (99.4%)	0.66
108	136Q	Q (100%)	130W	W (85.1%)	0.57
112	140F	F (95.6%)	134T	T (56%), V (26.3%)	0.67
119	147G	G (84.2%)	141V	V (73.7%)	0.62
133	161E	E (94.9%)	155I	I (44.6%), M (34.9%)	0.66
134	162A	A (90.5%)	156E	E (90.9%)	0.62
155	183P	P (95.6%)	177V	V (66.9%), I (30.3%)	0.66
167	195A	A (100%)	189S	S (94.9%)	0.57
172	200E	E (96.8%)	196G	G (53.1%)	0.66
251	299H	H (90.5%)	286P	P (61.7%)	0.62
353	410Q	Q (100%)	401S	S (75.4%)	0.67
368	426V	I (75.3%)	417N	N (92%)	0.64
381	440L	I (65.2%)	432F	F (78.3%)	0.60
384	443M	M (96.8%)	434K	W (89.7%)	0.68
388	447R	R (96.2%)	439M	A (32.6%), N (30.9%)	0.66
392	451C	C (79.7%)	443N	A (44.6%), N (26.3%)	0.64
396	455Q	Q (100%)	447K	N (42.3%), E (21.1%)	0.64
406	465F	F (84.8%)	457W	W (96%)	0.61
428	519K	K (79.7%)	509R	R (100%)	0.56
442	533S	S (58.2%), T (39.9%)	523M	M (92.6%)	0.64
443	534H	H (68.4%)	524P	P (97.1%)	0.65
446	537V	V (93.7%)	527Q	Q (89.1%)	0.65
458	550L	L (60.1%), M (26.6%)	547V	F (84%)	0.59
459	551Y	Y (98.7%)	547I	I (92.6%)	0.67
460	552L	L (94.3%)	548A	A (30.3%), C (22.9%), V (21.1%)	0.63
466	558Y	H (79.7%)	554T	T (92.6%)	0.65
473	565S	S (100%)	561L	L (90.3%)	0.68
523	616I	I (76.6%), V (22.2%)	612A	A (78.9%)	0.62
526	619L	L (96.2%)	615W	W (88%)	0.52
527	620W	W (98.7%)	616A	Y (55.4%), F (29.1%)	0.65

The reviewer: the analysis and the structures determined in the absence of tRNA. This represents a caveat to the conclusions that the authors should consider. While we know that IleRS is not an ARS that requires tRNA for the first step, it might be worth asking whether or not the structural changes seen in the comparison of the Ile-AMS and mupirocin structure might be altered by the presence of tRNA (see Note 3 below). (It is appreciated that this might be speculative.) This question is linked to the relative conformational mobility of the IleRS1 and IleRS2 forms (Note 4). Crystallography can provide a “snapshot” of the structure(s) but dynamic information is lacking; this might have to be interrogated by molecular dynamics, as the authors report Supp Figure 12.

Note 3. Authors do not comment on the possibility that the presence of the tRNA might well affect these parameters. May not alter general conclusions, but might alter the specifics.

Note 4. Page 12, structural comparison: what does the crystallography imply about the mobility of these various regions?

Response: *We agree with the reviewer that whether and how tRNA modulates ligand binding (i.e. resistance) is relevant and should be addressed in the manuscript. Kinetic data show that the K_i for mupirocin measured in the activation step (i.e. in the absence of tRNA) is similar to the K_i measured in the two-step aminoacylation reaction that includes the tRNA, suggesting that tRNA does not have a significant impact (for example; yeast ScIleRS2 has K_i of 8.6 μ M in the activation and of 15 μ M in the aminoacylation while *E. coli* IleRS1 has K_i of 6 nM in the activation step and of 2.5 nM in the aminoacylation). In agreement with these kinetic data, an overlay of our PmIleRS1:mupirocin structure (Fig. 5B) and the previously deposited SalleRS1:mupirocin structure in complex with tRNA (1QUE) revealed similar conformations of the active site residues contacting mupirocin, indicating that tRNA does not significantly modify mupirocin binding.*

We found that the KMSKS loop has higher local temperature factors in our structures, which is consistent with the flexibility and capacity of this loop to adopt different conformations during the various reaction stages (Kobayashi et al, JMB, 2005, 346(1):105-17). The binding of the 3'-end of the tRNA^{Leu} in the editing or aminoacylation conformations to LeuRS induces different conformations of the KMSKS loop (Palencia et al, Nat Struct Mol Biol., 2012, 19(7):677-84.). The structural data on IleRS includes only one tRNA-bound structure in which the 3'end of the tRNA faces the editing domain (Silvian et al, Science, 1999, 285: 1074-1077) precluding insights into the conformation of the KMSKS loop adopted when the tRNA binds to the aminoacylation site.

To address this point, a paragraph is introduced into the revised version (Page 14, lines 317-328)

A possible role of tRNA in modulating the binding of mupirocin seems unlikely, as kinetic data showed that the K_i for mupirocin in the amino acid activation and the two-step tRNA aminoacylation reactions are similar^{17,18,36}. Consistent with these findings, no direct interaction between mupirocin and the tRNA was observed in the structure of SalleRS1 complexed to tRNA and mupirocin (PDB 1QU2)¹⁴. Further, the active site residues contacting mupirocin in our PmIleRS1:mupirocin structure (Fig 5B) and the SalleRS1:tRNA:mupirocin complex, including the KMSKS loop and the tip of the α 1-helix harbouring the HIGH motif, adopt a highly similar arrangement, indicating that their conformation is independent of the presence of tRNA. Nevertheless, based on our structural findings we cannot exclude that upon the binding of the 3'-end of the tRNA to the catalytic site, the KMSKS loop, which displays increased local temperature factors, may become directly or indirectly stabilized or change its conformation.³⁷

The reviewer: A third question concerns the phenomenon of “hyper-resistance”. As yet, the authors base their conclusions on this point on modelling experiments; query whether actual experimental support is needed.

Response: Our structure of the hyper-resistant mut-GVHH-PmlleRS2 revealed that mupirocin would be prevented from binding to the hyper-resistant enzyme due to steric hindrance. Thus, it is not surprising that our efforts to obtain crystals for mut-GVHH-PmlleRS2 bound to mupirocin were unsuccessful, even when the inhibitor was used at very high concentrations (>10mM). Hence, the only reasonable way to visualize mupirocin binding to a hyper-resistant enzyme was by modelling.

The reviewer: Note 5 On page 22, the authors write, “in bacteria two lleRS types, lleRS1 and lleRS2 alternatively perform the housekeeping function”. How strongly has this been established? In the absence of Mup selection, can bacteria tolerate lleRS2 deletion without a loss of fitness?

Response: Both lleRS1 and lleRS2 can be sole housekeeping enzymes. Our wording might have been confusing and is clarified now by changing the sentence (page 13, Line 294) “In bacteria two lleRS types, lleRS1 and lleRS2, alternatively perform the housekeeping function” into

In bacteria, two lleRS types, lleRS1 and lleRS2, may perform the housekeeping function.

Regarding the question related to how lleRS1 and/or lleRS2 may support bacterial fitness, we have recently shown that lleRS2 enzymes perform a housekeeping role mainly in slower-growing bacteria (Zanki et al, Pro Sci. 2022;31:e4418). In sharp contrast, faster-growing bacteria rely on lleRS1 (sensitive) and sometimes also carry lleRS2 for resistance. We also found that lleRS2 cannot support the same speed of translation as lleRS1, which likely explains the lleRS distribution. It is plausible to assume that the requirement for keeping different translational rates might have resulted in different evolutionary pressures for lleRS1 and lleRS2. This point was commented on in the last paragraph of the discussion in the initially submitted version and is kept in the revised version.

RESPONSES TO REVIEWER 2:

The reviewer: There is only one suggestion to the authors, which is to amplify/clarify the discussion at the top of page 5. For clarity and biological insight, it would be useful in this section to provide information about the organism that makes pseudomonic acid (mupirocin) with particular reference to the organisms from which the crystal structures were derived. It is confusing to sort out the differences between the lleRS2 versions with a canonical and those with a non-canonical HXGH signature.

Response: Related to the reviewer’s comments we have introduced two changes in the introduction section.

i) at Page 3, lines 61-62 we added the organism which produces mupirocin and included references to the known structural data. A detailed description of the previously published structural data remained at the end of this paragraph.

ii) we changed the wording in the last paragraph of the introduction to better introduce the altered signature motif present in some lleRS2. The novel text is:

We found that some lleRS2 harbor an altered Class I HXGH signature motif (with X representing a hydrophobic residue) such that the 1st and the 3rd amino acids are swapped. This GXHH altered signature motif conveys lleRS2 with hyper-resistance to mupirocin, while

catalytic activity is only mildly affected. We determined structures from Priestia (Bacillus) megaterium²³ wild-type IleRS1 and IleRS2, both carrying the canonical signature motif, as well as mutants with a swapped GXHH motif, complexed to an aminoacyl-adenylate analog or mupirocin (Page 4, lines 80-86)

RESPONSES TO REVIEWER 3:

The reviewer: However, there are still some minor issues that need to be resolved before publication. I recommend replication experiments be conducted as the results of single molecule dynamics simulations can be misleading, the general rule is that at least three replicates should be performed to verify the reliability and stability of simulation results.

Response: *We agree with the reviewer and performed MD simulations in triplicates (originally we had duplicates). No changes in the outcome of the MD simulation experiments have been observed. The trajectories of triplicates are now presented in the revised version of Supplementary information in Supplementary Fig. 12 and the novel Supplementary Fig. 13.*

REVIEWERS' COMMENTS

Reviewer #1 (Remarks to the Author):

the revised manuscript does a very job of responding to all of the issues raised during initial review. I would agree that the supplementary bioinformatics analysis performed is not essential to include in the ms, because none of the conclusions of the work were altered. There are no further issues remaining. Congratulations on an excellent story.